# REPRESENTATIONS OF COMPUTER PROGRAMS IN THE HUMAN BRAIN

## ABSTRACT

We present the first study relating representations of computer programs generated by unsupervised machine learning (ML) models and representations of computer programs in the human brain. We analyze recordings—brain representations—from functional magnetic resonance imaging (fMRI) studies of people comprehending Python code. We discover brain representations, in different and specific regions of the brain, that encode static and dynamic properties of code such as abstract syntax tree (AST)-related information and runtime information. We also map brain representations to representations of a suite of ML models that vary in their complexity. We find that the Multiple Demand system, a system of brain regions previously shown to respond to code, contains information about multiple specific code properties, as well as machine learned representations of code. We make all the corresponding code, data, and analysis publicly available.[1]

## 1 INTRODUCTION

A question of interest, as yet uninvestigated, is what aspects of computer programs, *i.e.* code, are encoded by the human brain when comprehending them. Is it possible that common code properties and the semantics of programs are encoded in brain activity patterns when code is comprehended? A few prior works have investigated the neural bases of programming (Siegmund et al., 2017; Floyd et al., 2017; Peitek et al., 2018; Castelhano et al., 2019; Huang et al., 2019; Krueger et al., 2020; Prat et al., 2020; Ivanova et al., 2020; Liu et al., 2020; Ikutani et al., 2021; Peitek et al., 2021). These works use analysis techniques like functional magnetic resonance imaging (fMRI) and electroencephalography (EEG) to investigate brain regions involved in coding tasks like code comprehension, code writing, and data structure manipulation. The primary quest of these works though has been to locate physical regions in the brain involved in these activities, with the goal of determining whether code-related activity joins other activities supported by those brain regions. It remains unclear what specific code-related information these regions encode. Do these regions encode specific syntactic or semantic code properties? Further, are different regions involved in encoding different such properties? To date, no systematic method exists to generate the evidence from which to draw answers. These questions form the focus of this work.

One way to learn what information is encoded in the brain is by recording brain signals when reading code (through fMRI or EEG), and then decoding from the recorded signals a code property of interest. Being able to decode the property accurately from a specific region of the brain establishes that information related to that code property is faithfully represented in that brain region. A question central to such a decoding analysis would be the choice of the target code property–what code properties should be investigated? We can hand-select a set of fundamental properties of code and test if they can be decoded. While helpful, such a set will not preclude other, more complex aspects of code being encoded.

To address the limitation of hand-selected properties, we look to recent advances in unsupervised machine learning (ML) models trained on code. Dubbed *code models*, they process large corpora of code to learn *ML model representations* of computer programs. These models are increasingly being used in software engineering workflows (Allamanis et al., 2018a), and have been shown to perform well on tasks like code summarization (Alon et al., 2019), detecting variable misuse (Bichsel et al., 2016), and more recently, code auto-completion (Chen et al., 2021). These continuous representations likely encode syntax and semantics of programs (Bichsel et al., 2016; Allamanis et al., 2018b; Srikant et al., 2021). Beyond the fundamental, hand-selected properties, they may encode and describe more complex code properties. Decoding these continuous representations from brain activity data then promises to inform us whether such additional properties are also encoded

---

[1]Project URL - `https://github.com/anonmyous-author/anonymous-code`

in the brain. Further, comparing representations from code models to the code-related information encoded in the brain also informs us whether code models resemble humans in the knowledge they learn and encode. Yamins et al. (2013) first showed how information encoded in our visual system resembles what convolutional neural networks learn when trained to recognize images. A similar correspondence can possibly be established between code models and the human brain.

In this work, we study code comprehension, and utilize the fMRI recordings dataset from Ivanova et al. (2020) to investigate human brain representations of code. We present two means of proceeding—probing of brain region representations for specific code properties, and analyzing the mapping of these representations onto various code models with differing model complexity. We learn affine maps from brain representations to predict hand-selected code properties which summarize the syntactic and semantic behavior of programs, and similarly predict code representations of different code models. We investigate the effects that brain regions, the nature of code properties, and the complexity of the code models have on the accuracies with which we decode brain representations. We find that we can decode code-related information from brain representations, and can map representations of code models from brain representations. We find the Multiple Demand system, followed by the Language system, to consistently encode specific code properties and map with machine learned representations of code. We provide an open-source framework to replicate our experiments, and we release our data and analysis publicly. Link - `https://github.com/anonmyous-author/anonymous-code`. The framework allows both brain data and program metrics to be easily added to extend the current set of experiments. This should enable authors from other neuroimaging studies or code model developers to collaborate and analyze data across these works, which will also help amortize the high costs of carrying out such experiments.

## 2 RELATED WORK

Of the prior works that have investigated the neural bases of programming through fMRI and EEG techniques (Siegmund et al., 2017; Floyd et al., 2017; Peitek et al., 2018; Castelhano et al., 2019; Huang et al., 2019; Krueger et al., 2020; Ivanova et al., 2020; Liu et al., 2020; Ikutani et al., 2021; Peitek et al., 2021) and through behavioral studies (Prat et al., 2020; Casalnuovo et al., 2020; Crichton et al., 2021), the following probe brain recordings for program properties encoded in them.

Floyd et al. (2017) learn a linear model to successfully classify whether an observed brain activity corresponds to reading code or reading text. Ikutani et al. (2021) study expert programmers and show that it is possible to classify code into the four problem categories–math, search, sort, and string from the brain activations corresponding to the code. Similarly, Liu et al. (2020) classify whether a brain signal corresponds to code implementing an `if` condition or not. Peitek et al. (2021) analyze correlations between brain recordings of participants reading code and a set of code complexity metrics.

In testing for code properties, our work uses a similar methodology (a linear model trained on fMRI data), but we evaluate a larger set of static and dynamic code properties. We also systematically test whether key programming constructs like control flow and data operations are encoded by these brain representations. A systematic analysis of multiple such properties has not been performed in any of these works. Further, we perform these tests in those brain regions identified by Liu et al. (2020) and Ivanova et al. (2020) as being responsive specifically to code comprehension. This gives us a finer insight into how different regions encode these properties. In addition to these tests, we study representations generated by a suite of ML models with varying model complexity and compare those representations to brain representations.

Brain representations have also been studied in domains like natural language, vision, and motor control. Among related works in natural language, a domain that resembles programming languages, Mitchell et al. (2008); Pallier et al. (2011); Brennan & Pylkkänen (2017); Jain & Huth (2018); Gauthier & Levy (2019); Schwartz et al. (2019); Wang et al. (2020); Schrimpf et al. (2020); Cao et al. (2021) have studied brain representations of words and sentences by relating them to representations produced by language models. While the broader tools we use to investigate these representations, like multi-voxel pattern analysis (MVPA), are similar to some of these prior works, our focus is on properties specific to code and not natural language.

## 3 BACKGROUND

We provide a brief background on fMRI signals as a proxy for brain representations and describe the brain systems that we probe in this work.

**Measuring brain activity with fMRI.** Functional magnetic resonance imaging (fMRI) is a brain imaging technique used to measure brain activity in specific brain regions. When a brain region is active, blood flows into the region to aid its processing. An MRI machine measures this change in blood flow, and reports BOLD (blood oxygen level dependent) values sampled at the machine's frequency (Glover, 2011, usually 2 seconds). The smallest unit of brain tissue for which BOLD signal is recorded is called a voxel (an equivalent of a 3D pixel); it comprises several cubic millimeters of brain tissue. For our analyses, we select subsets of voxels belonging to specific brain systems. Following common practice, the parameters of a general linear model, fit to time-varying BOLD values, are used as a measure of the overall activation in each voxel in response to a given input. It is the values of these parameters that, according the neuroscience community, are brain representations.

**Brain systems.** A system of brain regions can span different areas of the brain but behaves as a holistic unit, showing similar patterns of engagement across a given cognitive task. We probe the following systems of regions in our work. See Appendix A for a detailed description of these systems along with relevant references.

Figure 1: The approximate locations of MD and the Language systems in the human brain. The regions depicted are used as a starting point to functionally localize these systems in individual participants.

• **Multiple Demand (MD) system.** This system of regions is known to engage in cognitively demanding, domain agnostic tasks like general problem solving, logic, and spatial memory tasks. Liu et al. (2020) and Ivanova et al. (2020) reported that this system is active during code comprehension.

• **Language system (LS).** This system responds during language production and comprehension across modalities (speech, text) and languages (across 10 language families, including American sign language). Siegmund et al. (2014; 2017) reported activity in regions resembling the LS during code comprehension, whereas Floyd et al. (2017); Liu et al. (2020); Krueger et al. (2020) concluded that brain regions processing language and code are distinct. Ivanova et al. (2020) reported moderate levels of activity within the LS in response to Python, but not to code in a visual programming language, ScratchJr.

We also probe brain representations in two brain systems responsible for perception.

• **Visual system (VS).** These regions respond primarily to visual inputs. We expect activity in this system to reflect low-level visual properties of the code (*e.g.* code length and indentation).

• **Auditory system.** These regions respond primarily to auditory inputs. We do not expect activity in this system to reflect code-related properties.

## 4 BRAIN AND MODEL REPRESENTATIONS

We describe in this section the method we follow to gather representations of code in the brain (Section 4.1), evaluate the different code properties they encode (Section 4.2), and how we compare brain representations to those generated by ML models (Section 4.3).

### 4.1 BRAIN REPRESENTATIONS AND DECODING

We provide a summary of how we process activation signals in the brain elicited by code comprehension, to probe whether they encode any specific code properties. We provide details in Appendix B.

**Dataset.** We use the publicly available brain recordings released as part of the study by Ivanova et al. (2020). It contains brain recordings of 24 participants reading 72 programs from a set of 108 unique Python programs. The 72 programs were presented in 12 blocks of 6 programs each. These programs were 3-10 lines in length and contained simple Python constructs, such as lists, *for* loops and *if* statements. A whole program was presented at once, and the task required participants to read the code and mentally compute the expected output, press a button when done, and select one of four choices presented to them which matched their calculated output.

**From dataset to brain representations.** The original dataset contains 3D images of the brain of each participant. Each voxel value in these images is an estimate of the response strength in this voxel when a particular code (or sentence) problem is presented. To determine which brain systems contain information about particular code properties, we focus our analyses to four systems – MD, Language, Visual, and Auditory (Section 3). A vector of voxels' activation values in each brain system is then taken to constitute that system's *representation* of a computer program and serves as an input to all our analyses. See Appendix B for details on data processing and voxel selection.

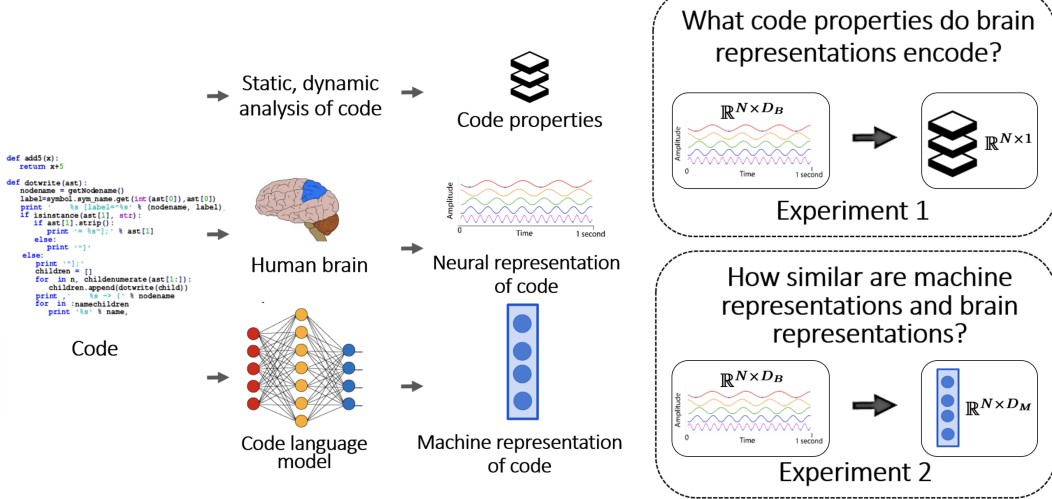

Figure 2: **Overview.** The goal of this work is to relate brain representations of code to (1) specific code properties and (2) representations of code produced by language models trained on code. In Experiment 1, we predict the different static and dynamic analysis metrics from the brain MRI recordings (each of dimension $D_B$) of 24 human subjects reading 72 unique Python programs $(N)$ by training separate linear models for each subject and metric. In Experiment 2, we learn affine maps from brain representations to the corresponding representations generated by code language models (each of dimension $D_M$) on these 72 programs.

**Analyzing brain representations of code.** We probe brain representations from each participant separately. We do not average data across participants since the regions which respond to any task (comprehending code in our case) need not align anatomically. For each of two experiments—decoding different code properties, and mapping to code model representations—we train ridge regression/classification models which take as input normalized brain representations per participant. We hence learn 24 different regression models, for each code property or code model (one per participant), and then report the mean performance of these models across participants. This procedure is also referred to as multi-voxel pattern analysis (MVPA) (Norman et al., 2006). Linear models are conventionally preferred for probes into brain representations since there has been evidence supporting the idea that other brain areas linearly map information from such brain representations (Kamitani & Tong, 2005; Kriegeskorte, 2011). We choose a linear model primarily to control for over-fitting in light of the relatively small dataset.

**A remark on data scarcity.** We train a linear regression/classification model with L2-regularization for every participant on unique cross-validated leave-one-run-out folds of the 72 programs they attempted (or 48 programs when sentences are removed). On the order of 1000 voxels were selected from each brain region responding to any given program per participant. As a baseline for the model predictions, we use the accuracy of a null permutation distribution generated from sampling 1000 random assignments of the labels.

## 4.2 CODE PROPERTIES

We attempt to decode the following code properties from brain representations.
• **Code vs. sentences.** The dataset provides a control condition where code problems are described in English sentences (referred to as *sentences*). See Figure B, Appendix B for an example. We classify these two stimuli categories from brain data.
• **Variable language.** We classify programs containing meaningful variables names from brain representations. The dataset has half its programs with variables in English, and the other half in Japanese (written in English characters), which do not convey any contextual meaning.
• **Control flow.** We predict whether a program contains a loop (`for` loop), a branch (`if` condition), or has sequential instructions without any control branching. The dataset contains an equal number of programs with each of the three conditions.
• **Data type.** We predict whether a program contains string operations or numeric operations in them. Half the programs in the dataset exclusively have string operations within them, while the other half are exclusively numeric.
• **Static analysis.** We decode static properties of a program. Specifically, we predict *token count* (number of tokens in the program) and *node count* (number of AST nodes). We also consider *cyclomatic complexity*, and *Halstead difficulty* (details in Appendix B), which have been used by software engineering practitioners to quantify the complexity of code, and to quantify the difficulty a human

would experience when comprehending code respectively. We defer predicting other advanced static analysis metrics such as tracking abstract interpretation joins, data flow analysis-related metrics, *etc*. to future work.

• ***Dynamic analysis.*** We decode information about a code's execution behavior. We evaluate two properties–*runtime steps* (number of instructions executed in the program) and *bytecode ops* (number of bytecode operations executed in running the program). We expect these properties to be correlated with the number of *mental instructions* needed to arrive at the output.

**Program length as a potential confound.** Program length is a potential confound in our experiments because the properties we examine can also potentially be differentiated using program length and other low-level code features. We measured the inter-correlations of these properties, and their correlation to the number of tokens in the program (program length) (see Appendix G). We expectedly found the four *static analysis* properties to be highly correlated to each other and to *bytecode ops*. We hence use one representative metric each from the two categories of properties for the rest of our analysis–*token count* for *static analysis*, and *runtime steps* for *dynamic analysis*. Importantly, the other properties we examine cannot be explained by program length alone, and therefore program length is not a confound in our experiments.

The brain representations (Section 4.1) are mapped to each of the code properties by training a ridge regression or classification model each for every participant-property pair. To evaluate model performance, we use classification accuracy when the predicted values are categorical (*e.g.* string vs. numeric data types), and the Pearson correlation coefficient when the predicted values are continuous (*e.g.* number of runtime steps). We choose Pearson correlation over RMSE, the canonical distance metric for continuous values, for its simplicity and interpretability. See Appendix I for the RMSE results, which are similar to the correlation results. When testing for the significance of these predictions, we perform false discovery rate (FDR) correction for the number of brain systems tested and the number of properties tested. See Appendix B for a detailed description of model hyper-parameters and cross-validation settings employed.

### 4.3 MODEL REPRESENTATIONS AND DECODING.

We evaluate a bench of unsupervised language models, spanning from count-based language models to transformer neural networks (Vaswani et al., 2017). These models were all trained on large (∼1M programs) Python datasets (Husain et al., 2019; Puri et al., 2021). We use the output of the trained encoders (raw logits) in each of the neural network models as representations of the code input to the model. We vary the general complexity of these models to test whether that variation is meaningful in establishing the quality of brain to model fits. Model complexity here is the number of a model's learnable parameters. We evaluate the following models, ordered by their increasing model complexity: simple frequency-based language models—*bag-of-words*, *TF-IDF*; auto-encoder based unsupervised models—*seq2seq* (Sutskever et al., 2014), *CodeTransformer* (Zügner et al., 2021), *CodeBERTa* (HuggingFace, 2020); auto-regressive model with a model complexity similar to that of transformers and *CodeBERTa—XLNet* (Yang et al., 2019).

We compare the results of the above models against an aggressive baseline (relative to the null-distribution labeling baseline), a *token projection model* provided by using unique Gaussian-distributed random vectors for the token embeddings in a vocabulary, and returning the sum of these token embeddings across a program. The resultant embedding is not transformed by any model or any weights–it instead serves as a proxy for the tokens that appear in the program. The results of this baseline model should be interpreted as the level of performance achievable from the presence of tokens alone with no structural information.

The brain representations (Section 4.1) are mapped to code representations by training another set of ridge regression models to learn an affine map, and a ranked accuracy metric is used to compare outputs. Ranked accuracy scores are commonly used in information retrieval where several elements in a range are similar to the correct one. In our case, the top-ranked prediction by the linear model indicates the closest fit (Euclidean distance) to the code model's representation. When reporting result significance, we perform false discovery rate (FDR) correction for the number of brain systems and the number of models. See Appendix B for a detailed discussion of the implemented code models and the corresponding metrics.

## 5 EXPERIMENTS & RESULTS

Our experiments address two research questions:
• **Experiment 1.** How well do the different brain systems encode specific code properties?
• **Experiment 2.** How do the brain representations of code correspond to the representations learned by computational language models of code?

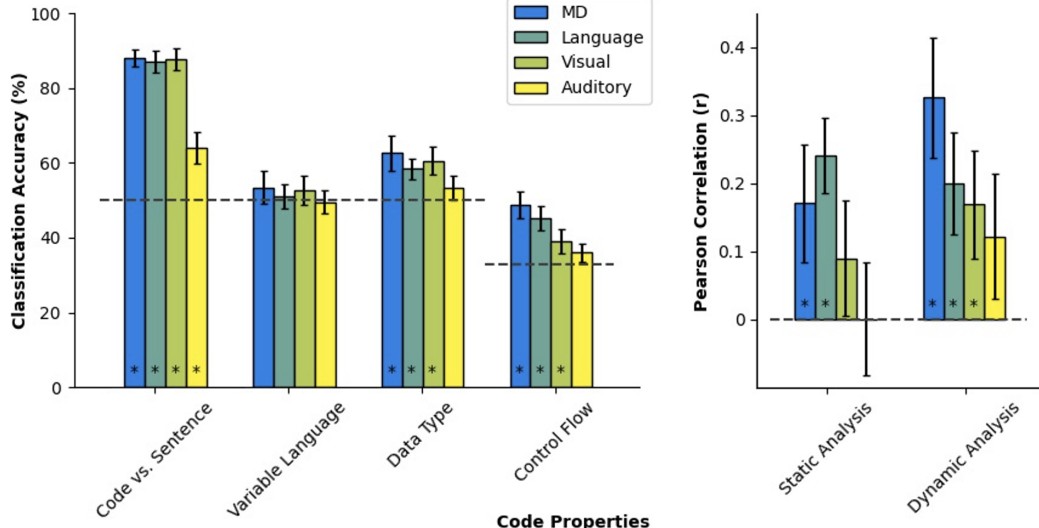

Figure 3: In Experiment 1, a linear model is trained on brain representations to predict each of the code properties described in Section 4.2. Error bars represent 95% confidence interval of individual subject scores. Dotted lines signify the empirical baseline for a null permutation distribution on shuffled labels. All results were compared to this permuted null distribution, and p-values for the number of comparisons in this experiment were corrected for false discovery rates (FDR). Statistically significant results are denoted with a ∗, marked at the base of the bars.

## 5.1 EXPERIMENT 1 - HOW WELL DO THE DIFFERENT BRAIN SYSTEMS ENCODE SPECIFIC CODE PROPERTIES?

Here, we analyze the classification models and regressions trained on brain representations to predict each of the code properties described in Section 4.2. The results of our analyses are summarized in Figure 3. The classification and regression tests are marked on the x-axis of the left and right subplots respectively; the classification accuracy or Pearson correlation for each of the tasks is marked on the y-axes. We plot dynamic and static properties separately from the others because their baselines are different due to a difference in the similarity metric (classification accuracy vs. Pearson correlation). The baselines for the categorical code properties differ from each other due to variation in the number of target classes.

**Auditory and Visual systems.** The Auditory system serves as a negative control for the other systems. Given that the code comprehension task we use is visual, we do not expect to decode any meaningful information from it. We find that, out of 6 tested properties, only *code vs sentences* can be decoded from the Auditory system, and even then the decoding performance is much lower than in other systems. The Visual system also serves as a control for low-level visual properties of the code (as opposed to abstract semantic or syntactic features). Given the visual attributes of a program such as the layout and indentation of the code, the length of the program, and the presence of letters and alphabets in the programs (Park et al., 2012; Roux et al., 2008; Polk et al., 2002), this system might reflect at least some of the properties we evaluate. This is indeed what we observe: 4 out of 6 tested properties can be decoded from the visual system above-chance. The MD and the Language systems yield the following specific observations. In a follow-up analysis (Analysis 3), we investigate the extent to which the MD and the Language systems represent decodable information beyond that which is represented in the Visual system.

**Analysis 1 - How accurately are different properties predicted by MD and LS?** We analyze how accurately each of MD and LS predict all the properties we evaluate.
• *Code vs. sentences*    Code and sentence properties are decoded most accurately among all the properties we test. The MD, LS, and VS decode with ∼ 90% accuracy. The high accuracy from MD validates claims from previous works suggesting the involvement of these regions in code comprehension.
• *Variable language*    This requires distinguishing between sets of nearly identical programs, with the only variation being in the language of the variable names (English vs. Japanese). We hence expect the Language system will most accurately encode this information, since it is sensitive to the presence of new words. However, no such effect is observed–no brain system shows any significant decoding. This unexpected finding is however consistent with Ivanova et al. (2020), who show a lack of any significant difference in the aggregate neural activity between the two variations. Both suggest that these brain systems do not seem to rely on variable names as a meaningful feature.
• *Control flow* and *data type*    Both *control flow* and *data type* tasks are most accurately predicted by the MD, followed by the LS. In the case of *control flow*, the more accurate predictions by the MD

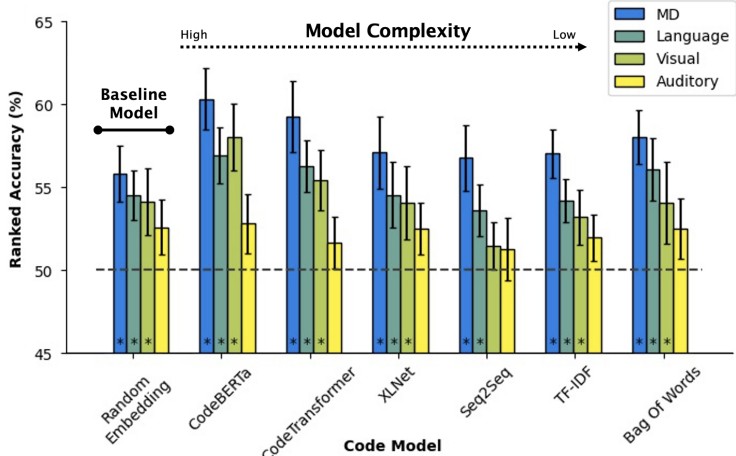

Figure 4: In Experiment 2, an affine map is learned from brain representations to the representations produced by machine learning models. Error bars represent 95% confidence interval of individual subject scores. Dotted lines signify the empirical baseline from a null permutation distribution on shuffled labels. *Random embedding* is an aggressive baseline of using random but unique embeddings for vocabulary tokens. Statistically significant results are denoted with a *, marked at the base of the bars.

can possibly be explained by the larger number of mental operations needed for programs with loops when compared to programs with one or no branches. The difference seen in *data type* prediction is unclear. While programs with string operations contain more literals and words from the English alphabet, we do not see any preference in the LS being able to decode this property more accurately. • *Static* and *Dynamic analysis* We observe the LS predicts *static analysis* properties with the highest accuracy. This is expected, since activity in the LS is known to be sensitive to the length of any text being read (independent of any specific program instructions in our case).
For the *dynamic analysis* property *runtime steps*, we see the MD system has the highest accuracy. This again is reflective of the workings of the MD system, which is sensitive to working memory tasks, and the number of steps executed by a program matches the number of mental calculations performed by a person tracing through the program to calculate its output.

**Analysis 2 - How accurately do MD and LS encode different properties?** We use t-tests to examine whether for a given property, any one brain system decodes it significantly more than another. We find no differences between the MD system and the Language system for any properties (Table 8, Appendix E). We additionally test if any brain system has a preference for a specific code property over another. For instance, is the evidence seen in Figure 3, of MD more accurately decoding *dynamic analysis* properties than *static analysis* properties, statistically significant? We do not find any significant differences (Table 9, Appendix E). While neither MD nor LS preferentially encode any of the code properties we explore in this work, we do find that the two brain systems individually encode these properties, which is a new result.

**Analysis 3 - Multi-system regression analysis.** The decoding performance of the Visual system is comparable to that of the MD and the Language systems (Figure 3). It is then possible that all three systems - MD, LS, and VS encode the same properties (all potentially related to program length and other low level program features). We employ a multi-system regression analysis to test this possibility. For each brain system, MD and LS ($S_i$), we train two models–one which decodes from $S_i$, and another which decodes from $S_i$+VS. If the difference in the prediction accuracies between the two models is significant, we conclude that $S_i$ encodes at least some information which is orthogonal to the information encoded by the Visual system. For *control flow*, *data type*, and *code vs sentences*, we find the MD to encode some information orthogonal to the VS. For *control flow* and *code vs sentences*, the LS also encodes some information orthogonal to what the VS does. This suggests that low-level code properties are insufficient to explain the key results from Experiment 1. Other combinations in the regression model show that the MD encodes information orthogonal to the LS when predicting *code vs sentences*. Detailed results are tabulated in Table 15, Appendix F.

### 5.2 EXPERIMENT 2 - HOW DO THE BRAIN REPRESENTATIONS OF CODE CORRESPOND TO THE REPRESENTATIONS LEARNED BY COMPUTATIONAL LANGUAGE MODELS OF CODE?

Here, we train ridge regression models with brain representations of programs from a specific brain system to predict code model representations of the same programs. The full set of results from this experiment are summarized in Figure 4.

**Auditory and Visual systems.** Similar to Experiment 1, the Auditory system performs as expected, exhibiting the lowest decoding performance across all code models. The VS exhibits the second

lowest performance across code models with the exception of *CodeBERTa* to which it maps more accurately to than LS, but not MD. However, this result is not statistically significant (Table 10, Appendix E).

**Analysis 1 - How well do brain representations in MD and LS map to code model representations?** To ease our analysis of the MD and LS systems, we re-plot in Figure 5 the decoding accuracies of just these two brain systems from Figure 4. We find that the MD system and Language system map to all the models in our suite significantly more accurately than the null permutation baseline (baseline accuracy of $50\%$ in Figures 4 and 5). Further, from Figure 5, we see the MD maps more accurately than the LS across all code models. We test the significance of the differences between the two systems and find that the MD ranked accuracy is higher than LS for *CodeBERTa*, *seq2seq*, and *TF-IDF*, while the differences are not significant for the other models (Table 10, Appendix E). We discuss the implication of these results in Section 6.

**Analysis 2 - The effect of model complexity on decoding to code models.** We investigate the impact model complexity has on the performance of the mapping between brain and code representations. Model complexity here is the number of a model's learnable parameters. In analysis 1 above, we saw that the MD system mapped to the representations of *CodeBERTa*, *seq2seq*, and *TF-IDF* significantly more accurately than the Language system. These three models vary substantially in their complexity, thus suggesting the lack of any relationship between model complexity and brain systems.

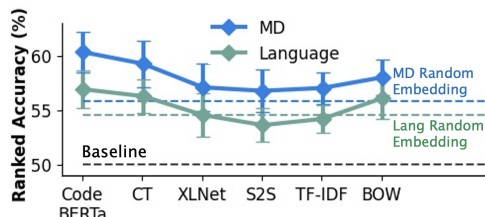

Figure 5: Mapping MD and LS to code model representations.

We also compare each of these code models against the *Random embedding* model. This baseline model assigns a random vector of numbers to every unique token that appears in the training vocabulary, and sums the vectors of all the tokens that appear in a program. By its design, this model encodes only the presence of specific tokens in programs. The mapping accuracies of random embeddings in the MD and the Language systems are marked in blue and green horizontal lines in Figure 5. We find that the Multiple Demand system maps to all the models more accurately than the *Random embedding*. This is not seen in the LS, which confirms the lack of variabilility of ranked accuracies observed in the LS.

In a set of pairwise significance tests, we find that the MD maps to *CodeBERTa* significantly more accurately than *Random embedding* while other systems are more accurate numerically, but not significantly (Table 11, Appendix E). We discuss its implications in the following section.

## 6 DISCUSSION

Through this study, we learn what computer program-related information can be decoded from the brain, and which brain systems primarily encode that information. In Experiment 1, we investigate whether a set of manually-selected code properties are encoded in different brain systems. In Experiment 2, by comparing brain representations to code model representations, we investigate if the brain encodes more than just the properties investigated in Experiment 1. Because the code models are trained on a much more expressive set of programs than the ones used in our Experiment 1, we also can learn what additional information these models learn about code.

Key among our contributions is demonstrating that it is possible to predict a range of code properties from brain system activity (representations). Experiment 1 shows that *control flow* and *data type* information, two fundamental program properties, as well as *dynamic analysis* and *static analysis* properties of code, can be decoded with high performance from the MD and the Language systems. In several cases, this decoding performance reflects decoding from information beyond what can be predicted from low-level features of the programs encoded in the Visual system alone. This is a new finding which indicates that the MD and the Language systems encode higher-level program-related information. Future work should additionally experiment with a larger set of programs that have more complex properties across multiple programming languages.

Another key contribution of our work, from Experiment 2, is demonstrating that it is possible to map brain representations to representations learned by code models. Unlike many prior decoding works in cognitive neuroscience, which record multiple responses to each of their experimental stimuli (done generally to control for the noise present in fMRI data), our decoders are trained on individual trials. This demonstrates the feasibility of probing aspects of code comprehension in the brain without the need to present the same program multiple times to a participant.

The performance of mappings between brain and code model representations does not appear to be correlated to the complexity of the models. However, we do observe a preferential encoding of the properties represented by code models in the MD over LS for the models *CodeBERTa*, *seq2seq*, and *TF-IDF*. Such a preferential encoding of properties in MD over LS was not seen for the properties we evaluated in Experiment 1, which provides evidence that the brain activation data encodes more than just the code properties we evaluated.

The MD and LS map to complex models like *seq2seq* and *CodeTransformer* almost as well as to a combination of random token embeddings. One plausible explanation for this surprising result is that the MD and LS signals we have access to mostly encode token-level information, and not richer structural information from the programs. The program stimuli are simple enough to allow the different properties evaluated in our work (*control flow*, *data type*, *etc.*) to be discerned from token level information alone (as validated in Experiment 1), which is likely why the random embeddings model is also able to predict these properties very well (see Table 4, Appendix C.2).

Taken together, this suggests that the information being decoded from brain activations in these two regions is driven at least by the information conveyed by tokens in the programs. We cannot come to this conclusion from the code properties investigated in Experiment 1 alone.

*CodeBERTa* alone, for which the response is significantly greater than random embeddings, and which maps more accurately to the MD system than any other brain system, provides initial evidence for the MD system encoding more than just token information. This will need more investigation. Given the current data and the number of observations we have, we can only conclude that tokens do get encoded well in both - brain activations and code models. This is a new finding on what aspects of code comprehension get encoded by our brains which can be successfully read out using a linear model.

**Limitations.** The average program in any software project exhibits non-trivial control and data dependencies, object manipulation, function calls, types, and state changes, which our dataset does not possess. However, the programming tasks in Ivanova et al. (2020) are short snippets of procedural code with limited program properties. Responses to longer, more realistic programs should be studied on a larger number of participants in the future, building on the understanding of simpler snippets provided by our work. Further, while there are equally important aspects to programming like designing solutions, selecting appropriate data structures, and writing programs, we have chosen to study a very specific activity related to programming—comprehending programs.

One aspect that cannot be inferred from these results, is whether the MD and LS are driven by the same underlying features of code that are used to discriminate between code properties, and to map to different code models. Future work should consider an encoding analysis, where we predict the activity of voxels in different brain systems from code properties, in order to establish the relative contributions of those properties to the activations of individual voxels in these systems.

**Broader impact.** Our findings have the potential to improve our understanding of the organization of the human brain, which can in turn lead to the design of better code models. In computer vision, recent results by Tschopp et al. (2018) show how deep network architectures that mimic the visual system in fruit flies exhibit superior image classification rates on image recognition tasks. One immediate outcome is reconsidering the current design of ML models of code. Extant code model architectures do not explicitly model the Multiple Demand system in any way–they only model syntactic information and infer dependency information from program ASTs. Taking inspiration from the role of the Multiple Demand system we identified in this work, modeling both static and dynamic information should be explored.

This work could perhaps also enhance *code prosthetics*–artificial interfaces in the body that can help the physically challenged engage with programming environments. Such systems generally rely on brain decoders–models that convert brain activity data to electrical impulses modulating external devices. An open challenge is to improve them. See the discussion in Nuyujukian et al. (2018) and Andersen et al. (2019) for details.

Finally, our aspiration is that the framework we release in this work will bring together the neuroscience and machine learning communities to better understand the cognitive and neural bases of programming.

## 7 REPRODUCABILITY STATEMENT

All the results, and corresponding tables, plots, and intermediate results we introduce in this paper are fully reproducible. We have made the source code to our work publicly available through this anonymized code repository - `https://github.com/anonmyous-author/anonymous-code/`

The repository contains detailed instructions for setting up the source code and running the two main experiments we introduce in this work. Further, a copy of the dataset released by Ivanova et al. Ivanova et al. (2020) is also available for in the repository, along with the source code.

All the intermediate results which we use to arrive at our conclusions are available in Appendix A through Appendix I.

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

# A BRAIN REGIONS

**Regions of Interest (ROIs).** We investigate the representations of code in well-studied systems of brain regions–the MD system, Language system, Visual system, and the Auditory system (details below). A *region* (also referred to as *parcel*) here denotes a contiguous chunk of brain mass involved in a cognitive task. A *system* of regions (also referred to as a *network*) can comprise multiple disjoint regions that exhibit shared activity patterns across a range of tasks. For many cognitive tasks, a region marks only the approximate location of voxel populations involved in a cognitive task. *Functional* ROIs (fROIs) are voxels within these broad regions that respond most strongly to a given task (language, working memory, *etc.*). The use of fROIs enables accounting for the exact anatomical locations of these task-sensitive voxels, which vary across individuals.

**Multiple Demand (MD) system.** Located in the prefrontal and parietal areas of the brain, this system of regions is active in a host of tasks requiring working memory and general problem solving skills, including math and logic (Duncan, 2010; Fedorenko et al., 2013; Amalric & Dehaene, 2019).

**Language system.** These regions have been identified to respond to both comprehension and production of language across modalities (written, speech, sign language), respond to typologically diverse languages (>50 languages, from across 10 language families), form a functionally integrated system, reliably and robustly track linguistic stimuli, and have been shown to be causally important for language (Clark & Cummings, 2003; Fedorenko et al., 2010; Blank & Fedorenko, 2017; Ayyash et al., 2021).

**Visual system.** These regions in the occipital lobe of the brain respond to visual input, ranging from low-level features like lines and edges, through intermediate categories like shapes, letters, and numbers, through higher order structures like faces and scenes. (Hubel & Wiesel, 1959; Polk et al., 2002; Epstein & Kanwisher, 1998; Kanwisher et al., 1997).

**Auditory system.** The auditory system is located in the superior temporal region of the brain. This region uniquely encodes pitch, speech, and music, but is not involved in high-level language comprehension and production (Norman-Haignere et al., 2015; 2019). In our experiments pertaining to programming language comprehension, we use the activity seen in the auditory system as a negative control.

## B  METHOD - DETAILS

**Selecting brain representations.** For each trial in the fMRI experiment, stimulus responses in each voxel were extracted from the parameters of a General Linear Model (GLM) fit to the time-varying BOLD signal in which each experimental condition was modeled with a boxcar function convolved with the canonical hemodynamic response function (HRF). For the localizer experiments, conditions were modeled as the entire block. For the Python program comprehension experiment, individual programs were modeled using the period from the onset of the code/sentence problem until the button press. The predictors for the GLM included trial ID (equivalent to problem ID), run number, and motion regressors. The voxels were then filtered using gray-matter masking and (for MD and the Language systems) network localization. Although fMRI measurements return whole brain responses, only a thin layer of cortex dubbed gray matter contains BOLD signal of interest to these analyses. Gray matter voxels were selected using a Bayesian segmentation of the anatomical brain image into standard tissue types, and then returning the set of indices where the posterior gray matter probability exceeds 0.70 (Ashburner et al., 1999). Next, these sets of voxels were filtered separately for each of the brain regions outlined in Appendix A. For the visual and auditory networks, primary sensory areas were identified using an anatomical atlas (Rolls et al., 2020). For the MD and the Language systems, voxels were functionally localized as those containing the top 10% of responses to their respective functional localizer tasks, as described in Ivanova et al. (2020). See Fedorenko et al. (2010) for a discussion of the functional localization approach as it pertains to the language network. GLM modeling and gray matter segmentation was performed using SPM12; functional localization was performed using the toolbox released by Ivanova et al. (2020). Once voxel responses within each brain region were extracted for each trial of each run, some additional preprocessing was required before finalizing the brain representations, and passing them to downstream models. Due to differences in MRI sensitivity across runs, each of the 12 runs of 6 programs for a given subject appeared with nonuniform mean and variance. In order to normalize these signals, so as to leverage data across a participant's entire scanning session, brain representations were z-transformed within each run to achieve a common scale. In order to avoid data-leakage from this preprocessing step, all downstream analyses were cross-validated via a leave-one-run-out approach, such that no intra-run rescaling could be used for prediction.

**MVPA cross-validation and hyperparameters.** For each brain system and each code property or code model, we run a separate MVPA analysis. For each subject, we train a separate ridge regression model on each cross-validated leave-one-run-out fold, and then predict the remaining targets using the brain representations from the left-out run. We use L2 regularization to control for model complexity, and employ a leave-one-out cross validation scheme to select $\lambda$. We score resulting predictions using a series of different metrics to maximize interpretability. For the categorical code properties, we report classification accuracy of model estimates.

For the continuously scaled code properties, we report Pearson correlation between the model estimates and true targets. For the model representations, we report rank accuracy between model representation predictions and the true model representations of left-out samples. We calculate rank accuracy as the percentile of a predicted and true target pairing within the full set of possible pairings sorted by a Euclidean distance metric. Rank accuracy was chosen for this experiment as it returns a more interpretable $50\%$ baseline through which to evaluate the brain to model mappings. Following metric calculation on each cross-validation fold, we take the mean of those estimates to report an out-of-sample score over the entire dataset for each subject. We then take the mean of those scores across subjects to derive an overall performance measure. As a baseline, we repeat the above process in its entirety 1000 times for each MVPA analysis, to provide a null permutation distribution against which to compare scores. These null distributions are fit with a univariate Gaussian and used to calculate z-statistics and p-values for MVPA scores. All statistical tests were corrected for false discovery rate. The code to run our MVPA analysis is available in `https://github.com/anonmyous-author/anonymous-code`.

**Code properties.** The code properties we consider are the following –
- **_Token count._** The total number of tokens that appear in the program. This is a sanity check to measure brain activity against, since it is natural to expect the activity in the brain to be correlated to the length of a program, and in general, the amount of content being comprehended. By their design, the dataset had program lengths with a small standard deviation.
- **_Node count._** The total number of AST nodes that appear in a program. Similar to the number of tokens, we verify if brain activity correlates to a proxy for the amount of syntactic content in it.
- **_Runtime steps._** We execute the programs and measure the number of instructions the program steps through. While two programs can have the same program length, the number of instructions executed can differ (*e.g.* `for 1:10` vs. `for 1:50`). We measure if any brain regions capture the number of *mental operations* needed to compute the output of a program.

- **Bytecode ops.** We execute the programs and count the number of bytecode operations performed. This metric should mimic the number of *mental operations* performed needed to compute the output of a program.
- **Cyclomatic complexity.** This metric (McCabe, 1976) is used to measure the general complexity of software systems. It is defined as a function of the number of nodes, edges, and the connected components in a program's control flow graph. While its efficacy as a metric to measure software complexity has been contested (Shepperd, 1988), we include it to see if brain activity is correlated to any explicit syntactic constructs of a program.
- **Halstead difficulty.** This metric (Halstead, 1977) was defined to measure how difficult any piece of software would be for a programmer to comprehend or write. It is defined as a function of the number of tokens, operations, vocabulary that appears in a program.

|  code  |  sentence  |
| --- | --- |
| height = 5
weight = 100
bmi = weight / (height*height)
print(bmi) | Your height is 5 feet and your weight is 100 pounds. The BMI is defined as the ratio between the weight and the square of the height of a person. What is your BMI? |

Figure 6: An example of a code problem and its sentence equivalent from Ivanova et al. (2020)

The inter-correlations between these metrics have been tabulated in Table 18, Appendix G.

**Code models - Model details.** We evaluate the following models in this work–

- **bag-of-words**, **TF-IDF**. These are count-based language models. They predict the likelihood of a token appearing in a program based on vocabulary statistics and frequencies.
- **seq2seq** (Sutskever et al., 2014), **CodeTransformer** (Zügner et al., 2021), **CodeBERTa** (HuggingFace, 2020). We evaluate these three autoencoder (AE) models with increasing model complexity. AE based pretraining reconstructs the original program from corrupted input.
- **XLNet** (Yang et al., 2019). We evaluate an auto-regressive (AR) model with a model complexity similar to that of transformers and *CodeBERTa*. AR language modeling estimates the probability distribution of a text corpus in one direction–either forward or backward, while AE models capture dependencies in both directions. We use an AR model to mimic a top-down, single pass comprehension style by humans.
- **Random embedding**. We compare the results of the above models against an aggressive baseline provided by using unique Gaussian-distributed random vectors for the token embeddings in a vocabulary, and returning the sum of these token embeddings across a program. The resultant embedding is not transformed by any model or any weights–it instead serves as a proxy for the tokens that appear in the program. This sets a higher bar than the null-distribution labeling baseline (Section 5.2).

**Code models - Configuration details.** In setting up our code model bench, we aimed to select a collection of models ranging in their complexity, namely *bag-of-words*, *TF-IDF* (Term frequency-Inverse document frequency), *seq2seq*, *XLNet*, *CodeTransformer*, and *CodeBERTa*, as well as a *Random embedding* embedding model.

- The *bag-of-words*, *TF-IDF*, *seq2seq*, and *Random embedding* models all use the same custom tokenizer defined by the current authors, whereas *XLNet*, *CodeTransformer*, and *CodeBERTa* use tokenizers defined by the original training authors (Zügner et al., 2021; HuggingFace, 2020).
- The tokenizer set up for the models trained by the current authors establishes a unique token in the vocabulary for each Python keyword, each Python builtin function, one token for all numeric types, one token for all string types, and $N$ tokens for each of the $N$ variables in a given program.
- In the case of the *Random embedding* baseline, this tokenizer was used to map each token in a piece of source code to an index in the vocabulary, which in turn was used to index a Gaussian distributed random matrix ($D = 128$). The sum of these random token projections was calculated to return a unique embedding for each program.
- For *bag-of-words* and *TF-IDF*, the validation split of the *code-search-net* Python dataset (Husain et al., 2019) was used to enumerate vocabulary and token occurrence statistics. These data were then used to transform each program to a vector where each dimension represented the raw (*bag-of-words*) or document-weighted (*TF-IDF*) count of the unique items in the vocabulary.
- If a given set of programs viewed by a subject never included a specific token, leading that column to equal the zero vector, then those dimensions would be filtered out.

- We used the dataset released by Project Codenet (Puri et al., 2021) to pretrain a *seq2seq* model (Sutskever et al., 2014). The dataset contains Python programs that are solutions to olympiad-style programming problems in data structures and algorithms. We trained a seq2seq model with a GRU unit for 15 epochs, with a dropout probability of 0.2, dot product attention, and a maximum sentence length of 500.
- For *XLNet*, *CodeTransformer*, and *CodeBERTa*, each program was passed through that model's tokenizer and model pipeline as defined by the original authors. The representations of each program were extracted from the pretrained encoders. *XLNet* and *CodeTransformer* were originally trained on the Python subset of the *code-search-net* dataset, whereas *CodeBERTa* was originally trained on the full *code-search-net* dataset, which also incorporates *go*, *java*, *javascript*, *php*, and *ruby*.

# C ADDITIONAL RESULTS

## C.1 EXPERIMENT 1

Using representations from localized brain regions, we attempt to decode static and dynamic properties of comprehended code, and learn maps to code model representations of that same code. We report decoding performance of each brain region to the original Ivanova et al. (2020) code properties in Table 1, the complexity-related code properties in Table 2, and the code model mappings in Table 3. We find that the MD system and the Language system decode all code properties and code models except *variable language* significantly above baseline, as established through a null permutation test. The Visual system decodes 4 of the 6 code properties and 6 of the 7 code models, whereas the Auditory system only decodes the *code vs sentences* property.

| Brain Representation Code Properties | Empirical Baseline | MD | Language | Visual | Auditory |
|---|---|---|---|---|---|
| Code vs. Sentence | 0.55 | 0.88 (+0.33) | 0.87 (+0.32) | 0.88 (+0.33) | 0.64 (+0.09) |
| Control Flow | 0.33 | 0.49 (+0.16) | 0.45 (+0.12) | 0.39 (+0.06) | 0.36 (+0.03) |
| Data Type | 0.49 | 0.63 (+0.14) | 0.58 (+0.09) | 0.61 (+0.12) | 0.53 (+0.04) |
| Variable Language | 0.50 | 0.53 (+0.03) | 0.51 (+0.01) | 0.53 (+0.03) | 0.50 (-0.00) |

Table 1: Brain region decoding performance on original Ivanova et al. (2020) code properties. Scores represent classification accuracy and are contrasted with an empirical baseline from the null permutation analysis. Values in parentheses are units above baseline.

| Brain Representation Code Properties | MD | Language | Visual | Auditory |
|---|---|---|---|---|
| Static Analysis | 0.17 | 0.24 | 0.09 | 0.00 |
| Dynamic Analysis | 0.33 | 0.20 | 0.17 | 0.12 |

Table 2: Brain region decoding performance on *static analysis* and *dynamic analysis* code properties. Scores represent Pearson correlation between predicted and true code properties.

## C.2 EXPERIMENT 2

| Brain Representation Code Models | Empirical Baseline | MD | Language | Visual | Auditory |
|---|---|---|---|---|---|
| Random Embedding | 0.50 | 0.56 (+0.06) | 0.55 (+0.05) | 0.54 (+0.04) | 0.53 (+0.03) |
| Bag Of Words | 0.50 | 0.58 (+0.08) | 0.56 (+0.06) | 0.54 (+0.04) | 0.52 (+0.02) |
| TF-IDF | 0.50 | 0.57 (+0.07) | 0.54 (+0.04) | 0.53 (+0.03) | 0.52 (+0.02) |
| Seq2Seq | 0.50 | 0.57 (+0.07) | 0.54 (+0.04) | 0.51 (+0.01) | 0.51 (+0.01) |
| XLNet | 0.50 | 0.57 (+0.07) | 0.55 (+0.05) | 0.54 (+0.04) | 0.52 (+0.02) |
| CodeTransformer | 0.50 | 0.59 (+0.09) | 0.56 (+0.06) | 0.55 (+0.05) | 0.52 (+0.02) |
| CodeBERTa | 0.50 | 0.60 (+0.10) | 0.57 (+0.07) | 0.58 (+0.08) | 0.53 (+0.03) |

Table 3: Brain region decoding performance on code model mappings. Scores represent rank accuracy and are contrasted with an empirical baseline from the null permutation analysis. Values in parentheses are units above baseline.

We additionally evaluate whether code models contain linearly decodable information about the code properties we explore in this work (Section 4.2). The models decode with near perfect accuracy on most classification tasks (avg. accuracy$= 0.96 \pm 0.01$), and with high correlations on the regression tasks (avg. $r = 0.89 \pm 0.02$). This is expected for the classification tasks, since the dataset contains tokens which help with perfectly separable decision boundaries. For example, for the *control flow* code property, the dataset contains programs with `if`, `for`, or neither, but never both. These results suggest that the code models we evaluate faithfully encode the set of properties we evaluate them on.

| Code Properties Empirical Baseline Model Representation | Control Flow 0.31 | Data Type 0.48 | Dynamic Analysis 0.00 | Static Analysis 0.00 |
|---|---|---|---|---|
| Random Embedding | 1.00 | 0.94 | 0.88 | 1.00 |
| Bag Of Words | 1.00 | 0.92 | 0.93 | 0.98 |
| TF-IDF | 1.00 | 0.94 | 0.94 | 0.89 |
| Seq2Seq | 0.95 | 0.97 | 0.87 | 0.91 |
| XLNet | 0.92 | 0.92 | 0.76 | 0.88 |
| CodeTransformer | 1.00 | 0.95 | 0.90 | 0.91 |
| CodeBERTa | 0.96 | 0.98 | 0.81 | 0.79 |

Table 4: Decoding performance of all models on all tasks. Scores represent classification accuracy for *control flow* and *data type*, and Pearson correlation for the remaining benchmarks. These can be contrasted with an empirical baseline from the null permutation analysis.

# D ANALYSIS OF VARIANCE

We run a series of one-way ANOVA statistical tests to assess whether decoding performance varies across brain regions for each code property and code model, and whether decoding performance varies across code models for each brain region. We report these results in Tables 5, 6, and 7. We find that decoding performance varies across brain region for all code properties and code models, except the *variable language* property and the *Random embedding* model, and decoding performance varies across code models for the MD system and the Visual system.

| Code Property | F | p | p (corrected) | Is Significant? |
|---|---|---|---|---|
| Code vs. Sentence | 53.57 | 4.03e-20 | 2.42e-19 | 1 |
| Control Flow | 12.52 | 6.13e-07 | 1.84e-06 | 1 |
| Static Analysis | 6.71 | 3.82e-04 | 7.65e-04 | 1 |
| Data Type | 4.46 | 5.66e-03 | 8.50e-03 | 1 |
| Dynamic Analysis | 4.15 | 8.38e-03 | 1.01e-02 | 1 |
| Variable Language | 0.82 | 4.84e-01 | 4.84e-01 | 0 |

Table 5: Results from statistical testing of variance across brain regions for each code property.

| Code Model | F | p | p (corrected) | Is Significant? |
|---|---|---|---|---|
| CodeTransformer | 11.85 | 1.25e-06 | 7.95e-06 | 1 |
| CodeBERTa | 11.28 | 2.27e-06 | 7.95e-06 | 1 |
| TF-IDF | 8.46 | 5.08e-05 | 9.48e-05 | 1 |
| Seq2Seq | 8.40 | 5.42e-05 | 9.48e-05 | 1 |
| Bag Of Words | 5.66 | 1.33e-03 | 1.86e-03 | 1 |
| XLNet | 3.48 | 1.90e-02 | 2.22e-02 | 1 |
| Random Embedding | 2.32 | 8.04e-02 | 8.04e-02 | 0 |

Table 6: Results from statistical testing of variance across brain regions for each code model.

| Brain Region | F | p | p (corrected) | Is Significant? |
|---|---|---|---|---|
| Visual | 4.00 | 9.17e-04 | 3.67e-03 | 1 |
| MD | 2.72 | 1.54e-02 | 3.08e-02 | 1 |
| Language | 2.18 | 4.70e-02 | 6.27e-02 | 0 |
| Auditory | 0.43 | 8.57e-01 | 8.57e-01 | 0 |

Table 7: Results from statistical testing of variance across code models for each brain region.

# E  PAIRWISE ANALYSIS

To extend the findings from the ANOVA analyses towards specific pairwise comparisons between brain regions and code models, we compare scores between brain regions for each code property or code model, and compare scores between code models and a subset of code properties for each brain region, using two-sample two-tail t-tests with unequal variance. We report all pairwise brain region comparisons in Tables 8 and 10, and report all pairwise code property and model comparisons in Tables 9 and 11. To highlight a few key results from code properties, we find that the MD system decodes *runtime steps* and *control flow* significantly better than the Visual system (Table 8). Additionally, the Language system decodes *static analysis* and *control flow* significantly better than the Visual system. Moving onto code models, we find that the MD system decodes *CodeBERTa*, *seq2seq*, and *TF-IDF* significantly more accurately than the Language system, and additionally decodes *CodeTransformer*, *seq2seq*, *TF-IDF*, and *bag-of-words* significantly more accurately than the Visual system (Table 10). Finally, an investigation into model complexity reveals that *CodeBERTa* is significantly more accurately decoded from the MD system than the *Random embedding* model.

| Code Property | Brain Region A | Brain Region B | t | p | p (corrected) | Is Significant? |
|---|---|---|---|---|---|---|
| Code vs. Sentence | Language | Auditory | 8.82 | 5.33e-11 | 6.40e-10 | 1 |
| Code vs. Sentence | MD | Auditory | 9.79 | 1.28e-11 | 4.62e-10 | 1 |
| Code vs. Sentence | Visual | Auditory | 9.06 | 2.65e-11 | 4.78e-10 | 1 |
| Control Flow | Language | Auditory | 4.28 | 1.02e-04 | 6.10e-04 | 1 |
| Control Flow | Language | Visual | 2.60 | 1.26e-02 | 3.30e-02 | 1 |
| Control Flow | MD | Auditory | 5.59 | 1.65e-06 | 1.49e-05 | 1 |
| Control Flow | MD | Visual | 3.92 | 2.95e-04 | 1.52e-03 | 1 |
| Data Type | MD | Auditory | 3.16 | 2.98e-03 | 1.21e-02 | 1 |
| Data Type | Visual | Auditory | 2.93 | 5.33e-03 | 1.88e-02 | 1 |
| Dynamic Analysis | MD | Auditory | 3.13 | 3.03e-03 | 1.21e-02 | 1 |
| Dynamic Analysis | MD | Visual | 2.59 | 1.28e-02 | 3.30e-02 | 1 |
| Static Analysis | Language | Auditory | 4.70 | 2.97e-05 | 2.14e-04 | 1 |
| Static Analysis | Language | Visual | 2.92 | 5.75e-03 | 1.88e-02 | 1 |
| Static Analysis | MD | Auditory | 2.79 | 7.68e-03 | 2.30e-02 | 1 |
| Code vs. Sentence | Language | Visual | -0.28 | 7.83e-01 | 8.30e-01 | 0 |
| Code vs. Sentence | MD | Language | 0.49 | 6.27e-01 | 6.84e-01 | 0 |
| Code vs. Sentence | MD | Visual | 0.18 | 8.55e-01 | 8.55e-01 | 0 |
| Control Flow | MD | Language | 1.44 | 1.58e-01 | 2.84e-01 | 0 |
| Control Flow | Visual | Auditory | 1.50 | 1.40e-01 | 2.81e-01 | 0 |
| Data Type | Language | Auditory | 2.30 | 2.62e-02 | 6.30e-02 | 0 |
| Data Type | Language | Visual | -0.94 | 3.55e-01 | 4.91e-01 | 0 |
| Data Type | MD | Language | 1.48 | 1.47e-01 | 2.81e-01 | 0 |
| Data Type | MD | Visual | 0.64 | 5.24e-01 | 6.18e-01 | 0 |
| Dynamic Analysis | Language | Auditory | 1.29 | 2.03e-01 | 3.04e-01 | 0 |
| Dynamic Analysis | Language | Visual | 0.57 | 5.73e-01 | 6.45e-01 | 0 |
| Dynamic Analysis | MD | Language | 2.11 | 4.01e-02 | 9.03e-02 | 0 |
| Dynamic Analysis | Visual | Auditory | 0.75 | 4.55e-01 | 5.85e-01 | 0 |
| Static Analysis | MD | Language | -1.34 | 1.89e-01 | 3.04e-01 | 0 |
| Static Analysis | MD | Visual | 1.32 | 1.94e-01 | 3.04e-01 | 0 |
| Static Analysis | Visual | Auditory | 1.47 | 1.48e-01 | 2.81e-01 | 0 |
| Variable Language | Language | Auditory | 0.66 | 5.15e-01 | 6.18e-01 | 0 |
| Variable Language | Language | Visual | -0.63 | 5.32e-01 | 6.18e-01 | 0 |
| Variable Language | MD | Auditory | 1.40 | 1.68e-01 | 2.88e-01 | 0 |
| Variable Language | MD | Language | 0.84 | 4.06e-01 | 5.41e-01 | 0 |
| Variable Language | MD | Visual | 0.23 | 8.20e-01 | 8.44e-01 | 0 |
| Variable Language | Visual | Auditory | 1.23 | 2.26e-01 | 3.25e-01 | 0 |

Table 8: Results from pairwise t-tests of brain regions for each code property. $+t$ reflects $A > B$, whereas $-t$ reflects $A < B$.

| Brain Region | Code Property A | Code Property B | t | p | p (corrected) | Is Significant? |
|---|---|---|---|---|---|---|
| Auditory | Static Analysis | Dynamic Analysis | -1.93 | 6.00e-02 | 1.20e-01 | 0 |
| Language | Static Analysis | Dynamic Analysis | 0.85 | 4.00e-01 | 4.00e-01 | 0 |
| MD | Static Analysis | Dynamic Analysis | -2.45 | 1.80e-02 | 7.18e-02 | 0 |
| Visual | Static Analysis | Dynamic Analysis | -1.34 | 1.88e-01 | 2.51e-01 | 0 |

Table 9: Results from pairwise t-tests of continuous code metrics across brain regions. Only this subset of properties was selected so as to evaluate scores with a consistent metric and baseline. $+t$ reflects $A > B$, whereas $-t$ reflects $A < B$.

| Code Model | Brain Region A | Brain Region B | t | p | p (corrected) | Is Significant? |
|---|---|---|---|---|---|---|
| Bag Of Words | Language | Auditory | 2.70 | 9.69e-03 | 2.60e-02 | 1 |
| Bag Of Words | MD | Auditory | 4.42 | 6.03e-05 | 6.33e-04 | 1 |
| Bag Of Words | MD | Visual | 2.61 | 1.26e-02 | 2.93e-02 | 1 |
| CodeBERTa | Language | Auditory | 3.29 | 1.94e-03 | 7.41e-03 | 1 |
| CodeBERTa | MD | Auditory | 5.80 | 5.85e-07 | 2.46e-05 | 1 |
| CodeBERTa | MD | Language | 2.67 | 1.05e-02 | 2.60e-02 | 1 |
| CodeBERTa | Visual | Auditory | 3.80 | 4.28e-04 | 2.25e-03 | 1 |
| CodeTransformer | Language | Auditory | 4.12 | 1.55e-04 | 1.09e-03 | 1 |
| CodeTransformer | MD | Auditory | 5.61 | 1.42e-06 | 2.99e-05 | 1 |
| CodeTransformer | MD | Visual | 2.67 | 1.05e-02 | 2.60e-02 | 1 |
| CodeTransformer | Visual | Auditory | 3.09 | 3.40e-03 | 1.19e-02 | 1 |
| Random Embedding | MD | Auditory | 2.69 | 9.95e-03 | 2.60e-02 | 1 |
| Seq2Seq | MD | Auditory | 3.94 | 2.72e-04 | 1.63e-03 | 1 |
| Seq2Seq | MD | Language | 2.48 | 1.71e-02 | 3.78e-02 | 1 |
| Seq2Seq | MD | Visual | 4.25 | 1.16e-04 | 9.71e-04 | 1 |
| TF-IDF | MD | Auditory | 4.95 | 1.06e-05 | 1.48e-04 | 1 |
| TF-IDF | MD | Language | 2.86 | 6.39e-03 | 2.06e-02 | 1 |
| TF-IDF | MD | Visual | 3.42 | 1.33e-03 | 6.20e-03 | 1 |
| XLNet | MD | Auditory | 3.39 | 1.53e-03 | 6.45e-03 | 1 |
| Bag Of Words | Language | Visual | 1.29 | 2.03e-01 | 2.80e-01 | 0 |
| Bag Of Words | MD | Language | 1.50 | 1.40e-01 | 2.03e-01 | 0 |
| Bag Of Words | Visual | Auditory | 0.98 | 3.33e-01 | 3.88e-01 | 0 |
| CodeBERTa | Language | Visual | -0.81 | 4.20e-01 | 4.65e-01 | 0 |
| CodeBERTa | MD | Visual | 1.66 | 1.04e-01 | 1.62e-01 | 0 |
| CodeTransformer | Language | Visual | 0.69 | 4.92e-01 | 5.30e-01 | 0 |
| CodeTransformer | MD | Language | 2.21 | 3.23e-02 | 6.46e-02 | 0 |
| Random Embedding | Language | Auditory | 1.70 | 9.56e-02 | 1.54e-01 | 0 |
| Random Embedding | Language | Visual | 0.32 | 7.52e-01 | 7.81e-01 | 0 |
| Random Embedding | MD | Language | 1.13 | 2.63e-01 | 3.18e-01 | 0 |
| Random Embedding | MD | Visual | 1.28 | 2.07e-01 | 2.80e-01 | 0 |
| Random Embedding | Visual | Auditory | 1.16 | 2.53e-01 | 3.18e-01 | 0 |
| Seq2Seq | Language | Auditory | 1.87 | 6.85e-02 | 1.20e-01 | 0 |
| Seq2Seq | Language | Visual | 1.98 | 5.36e-02 | 1.02e-01 | 0 |
| Seq2Seq | Visual | Auditory | 0.14 | 8.87e-01 | 8.87e-01 | 0 |
| TF-IDF | Language | Auditory | 2.28 | 2.71e-02 | 5.69e-02 | 0 |
| TF-IDF | Language | Visual | 0.92 | 3.64e-01 | 4.13e-01 | 0 |
| TF-IDF | Visual | Auditory | 1.13 | 2.65e-01 | 3.18e-01 | 0 |
| XLNet | Language | Auditory | 1.58 | 1.22e-01 | 1.83e-01 | 0 |
| XLNet | Language | Visual | 0.30 | 7.62e-01 | 7.81e-01 | 0 |
| XLNet | MD | Language | 1.71 | 9.49e-02 | 1.54e-01 | 0 |
| XLNet | MD | Visual | 1.91 | 6.25e-02 | 1.14e-01 | 0 |
| XLNet | Visual | Auditory | 1.13 | 2.64e-01 | 3.18e-01 | 0 |

Table 10: Results from pairwise t-tests of brain regions for each code model. $+t$ reflects $A > B$, whereas $-t$ reflects $A < B$.

| Brain Region | Code Model A | Code Model B | t | p | p (corrected) | Is Significant? |
|---|---|---|---|---|---|---|
| MD | Random Embedding | CodeBERTa | -3.53 | 9.65e-04 | 2.70e-02 | 1 |
| Visual | CodeBERTa | Seq2Seq | 5.13 | 7.01e-06 | 5.89e-04 | 1 |
| Visual | CodeBERTa | TF-IDF | 3.60 | 7.93e-04 | 2.70e-02 | 1 |
| Visual | CodeTransformer | Seq2Seq | 3.35 | 1.68e-03 | 3.53e-02 | 1 |
| Auditory | Bag Of Words | TF-IDF | 0.47 | 6.39e-01 | 8.14e-01 | 0 |
| Auditory | CodeBERTa | Bag Of Words | 0.23 | 8.23e-01 | 9.48e-01 | 0 |
| Auditory | CodeBERTa | CodeTransformer | 0.94 | 3.51e-01 | 6.82e-01 | 0 |
| Auditory | CodeBERTa | Seq2Seq | 1.15 | 2.58e-01 | 6.37e-01 | 0 |
| Auditory | CodeBERTa | TF-IDF | 0.73 | 4.68e-01 | 7.40e-01 | 0 |
| Auditory | CodeBERTa | XLNet | 0.24 | 8.10e-01 | 9.48e-01 | 0 |
| Auditory | CodeTransformer | Bag Of Words | -0.69 | 4.93e-01 | 7.40e-01 | 0 |
| Auditory | CodeTransformer | Seq2Seq | 0.30 | 7.64e-01 | 9.44e-01 | 0 |
| Auditory | CodeTransformer | TF-IDF | -0.27 | 7.87e-01 | 9.45e-01 | 0 |
| Auditory | CodeTransformer | XLNet | -0.75 | 4.57e-01 | 7.40e-01 | 0 |
| Auditory | Random Embedding | Bag Of Words | 0.07 | 9.45e-01 | 1.00e+00 | 0 |
| Auditory | Random Embedding | CodeBERTa | -0.16 | 8.70e-01 | 9.75e-01 | 0 |
| Auditory | Random Embedding | CodeTransformer | 0.80 | 4.29e-01 | 7.35e-01 | 0 |
| Auditory | Random Embedding | Seq2Seq | 1.02 | 3.14e-01 | 6.82e-01 | 0 |
| Auditory | Random Embedding | TF-IDF | 0.58 | 5.68e-01 | 7.74e-01 | 0 |
| Auditory | Random Embedding | XLNet | 0.08 | 9.40e-01 | 1.00e+00 | 0 |
| Auditory | Seq2Seq | Bag Of Words | -0.91 | 3.65e-01 | 6.82e-01 | 0 |
| Auditory | Seq2Seq | TF-IDF | -0.56 | 5.81e-01 | 7.74e-01 | 0 |
| Auditory | XLNet | Bag Of Words | 0.00 | 1.00e+00 | 1.00e+00 | 0 |
| Auditory | XLNet | Seq2Seq | 0.98 | 3.33e-01 | 6.82e-01 | 0 |
| Auditory | XLNet | TF-IDF | 0.52 | 6.07e-01 | 7.85e-01 | 0 |
| Language | Bag Of Words | TF-IDF | 1.64 | 1.10e-01 | 3.29e-01 | 0 |
| Language | CodeBERTa | Bag Of Words | 0.63 | 5.35e-01 | 7.72e-01 | 0 |
| Language | CodeBERTa | CodeTransformer | 0.54 | 5.90e-01 | 7.74e-01 | 0 |
| Language | CodeBERTa | Seq2Seq | 2.82 | 7.17e-03 | 1.15e-01 | 0 |
| Language | CodeBERTa | TF-IDF | 2.48 | 1.70e-02 | 1.33e-01 | 0 |
| Language | CodeBERTa | XLNet | 1.77 | 8.27e-02 | 2.89e-01 | 0 |
| Language | CodeTransformer | Bag Of Words | 0.14 | 8.89e-01 | 9.83e-01 | 0 |
| Language | CodeTransformer | Seq2Seq | 2.40 | 2.06e-02 | 1.33e-01 | 0 |
| Language | CodeTransformer | TF-IDF | 2.02 | 4.91e-02 | 2.37e-01 | 0 |
| Language | CodeTransformer | XLNet | 1.35 | 1.83e-01 | 4.81e-01 | 0 |
| Language | Random Embedding | Bag Of Words | -1.28 | 2.09e-01 | 5.32e-01 | 0 |
| Language | Random Embedding | CodeBERTa | -2.05 | 4.61e-02 | 2.37e-01 | 0 |
| Language | Random Embedding | CodeTransformer | -1.59 | 1.20e-01 | 3.47e-01 | 0 |
| Language | Random Embedding | Seq2Seq | 0.85 | 4.02e-01 | 7.03e-01 | 0 |
| Language | Random Embedding | TF-IDF | 0.34 | 7.33e-01 | 9.19e-01 | 0 |
| Language | Random Embedding | XLNet | 0.00 | 1.00e+00 | 1.00e+00 | 0 |
| Language | Seq2Seq | Bag Of Words | -2.01 | 5.08e-02 | 2.37e-01 | 0 |
| Language | Seq2Seq | TF-IDF | -0.56 | 5.77e-01 | 7.74e-01 | 0 |
| Language | XLNet | Bag Of Words | -1.12 | 2.69e-01 | 6.46e-01 | 0 |
| Language | XLNet | Seq2Seq | 0.72 | 4.75e-01 | 7.40e-01 | 0 |
| Language | XLNet | TF-IDF | 0.29 | 7.76e-01 | 9.45e-01 | 0 |
| MD | Bag Of Words | TF-IDF | 0.88 | 3.81e-01 | 6.82e-01 | 0 |
| MD | CodeBERTa | Bag Of Words | 1.84 | 7.18e-02 | 2.87e-01 | 0 |
| MD | CodeBERTa | CodeTransformer | 0.74 | 4.61e-01 | 7.40e-01 | 0 |
| MD | CodeBERTa | Seq2Seq | 2.60 | 1.27e-02 | 1.28e-01 | 0 |
| MD | CodeBERTa | TF-IDF | 2.77 | 8.21e-03 | 1.15e-01 | 0 |
| MD | CodeBERTa | XLNet | 2.25 | 2.95e-02 | 1.77e-01 | 0 |
| MD | CodeTransformer | Bag Of Words | 0.90 | 3.71e-01 | 6.82e-01 | 0 |
| MD | CodeTransformer | Seq2Seq | 1.68 | 1.00e-01 | 3.11e-01 | 0 |
| MD | CodeTransformer | TF-IDF | 1.69 | 9.84e-02 | 3.11e-01 | 0 |
| MD | CodeTransformer | XLNet | 1.40 | 1.67e-01 | 4.53e-01 | 0 |
| MD | Random Embedding | Bag Of Words | -1.81 | 7.67e-02 | 2.89e-01 | 0 |
| MD | Random Embedding | CodeTransformer | -2.46 | 1.81e-02 | 1.33e-01 | 0 |
| MD | Random Embedding | Seq2Seq | -0.70 | 4.86e-01 | 7.40e-01 | 0 |
| MD | Random Embedding | TF-IDF | -1.05 | 2.99e-01 | 6.82e-01 | 0 |
| MD | Random Embedding | XLNet | -0.89 | 3.76e-01 | 6.82e-01 | 0 |
| MD | Seq2Seq | Bag Of Words | -0.95 | 3.45e-01 | 6.82e-01 | 0 |
| MD | Seq2Seq | TF-IDF | -0.21 | 8.35e-01 | 9.48e-01 | 0 |
| MD | XLNet | Bag Of Words | -0.67 | 5.05e-01 | 7.44e-01 | 0 |
| MD | XLNet | Seq2Seq | 0.21 | 8.31e-01 | 9.48e-01 | 0 |
| MD | XLNet | TF-IDF | 0.04 | 9.65e-01 | 1.00e+00 | 0 |
| Visual | Bag Of Words | TF-IDF | 0.55 | 5.84e-01 | 7.74e-01 | 0 |
| Visual | CodeBERTa | Bag Of Words | 2.43 | 1.94e-02 | 1.33e-01 | 0 |
| Visual | CodeBERTa | CodeTransformer | 1.86 | 6.97e-02 | 2.87e-01 | 0 |
| Visual | CodeBERTa | XLNet | 2.56 | 1.38e-02 | 1.28e-01 | 0 |
| Visual | CodeTransformer | Bag Of Words | 0.89 | 3.80e-01 | 6.82e-01 | 0 |
| Visual | CodeTransformer | TF-IDF | 1.79 | 8.08e-02 | 2.89e-01 | 0 |
| Visual | CodeTransformer | XLNet | 0.93 | 3.57e-01 | 6.82e-01 | 0 |
| Visual | Random Embedding | Bag Of Words | 0.05 | 9.58e-01 | 1.00e+00 | 0 |
| Visual | Random Embedding | CodeBERTa | -2.67 | 1.04e-02 | 1.25e-01 | 0 |
| Visual | Random Embedding | CodeTransformer | -0.95 | 3.48e-01 | 6.82e-01 | 0 |
| Visual | Random Embedding | Seq2Seq | 2.11 | 4.07e-02 | 2.28e-01 | 0 |
| Visual | Random Embedding | TF-IDF | 0.70 | 4.87e-01 | 7.40e-01 | 0 |
| Visual | Random Embedding | XLNet | 0.04 | 9.70e-01 | 1.00e+00 | 0 |
| Visual | Seq2Seq | Bag Of Words | -1.76 | 8.74e-02 | 2.94e-01 | 0 |
| Visual | Seq2Seq | TF-IDF | -1.55 | 1.29e-01 | 3.60e-01 | 0 |
| Visual | XLNet | Bag Of Words | 0.02 | 9.86e-01 | 1.00e+00 | 0 |
| Visual | XLNet | Seq2Seq | 1.92 | 6.20e-02 | 2.74e-01 | 0 |
| Visual | XLNet | TF-IDF | 0.61 | 5.42e-01 | 7.72e-01 | 0 |

Table 11: Results from pairwise t-tests of code models for each brain region. $+t$ reflects $A > B$, whereas $-t$ reflects $A < B$.

## F    MULTI-SYSTEM REGRESSION ANALYSIS: SCORES AND SIGNIFICANCE

In order to investigate whether each of the brain systems included contribute unique information towards the decoding tasks, we combine brain representations from different systems in paired combinations and evaluate effects on downstream decoding performance across all experiments. We find that the addition of MD or LS to VS, relative to VS alone, improves downstream decoding of *code vs sentences*. The same effect is observed for the combination of MD and LS compared to either alone. These data suggest that the MD system and the Language system encode unique variance relevant to the decoding of *code vs sentences*, and this is above and beyond the information encoded in the Visual system or each other individually. Additionally, we observe that the addition of MD to VS, relative to VS alone, improves downstream decoding of *control flow* and *data type*. This same effect is observed for the addition of LS to Visual system for *control flow*. These data suggest that the MD system and the Language system encode unique information above and beyond information encoded in the Visual system for the *control flow* decoding task, and the MD system encodes unique information above and beyond information encoded in the Visual system for the *data type* decoding task (Table 15). We repeat this process for Experiment 2, where we find that the addition of MD to VS improves decoding for 5 models, the addition of MD to LS improves decoding for 4 models, and the addition of LS to Visual system improves decoding for 2 models (Table 16).

| Brain Representation | | L+V | MD+L | MD+V |
| Code Properties | Empirical Baseline | | | |
| --- | --- | --- | --- | --- |
| Code vs. Sentence | 0.56 | 0.94 (+0.38) | 0.94 (+0.38) | 0.93 (+0.37) |
| Control Flow | 0.33 | 0.45 (+0.12) | 0.49 (+0.16) | 0.49 (+0.16) |
| Data Type | 0.50 | 0.62 (+0.12) | 0.63 (+0.13) | 0.68 (+0.18) |
| Variable Language | 0.50 | 0.53 (+0.03) | 0.53 (+0.03) | 0.53 (+0.03) |

Table 12: Multi-system regression analysis on original Ivanova et al. (2020) code properties. Scores represent classification accuracy and are contrasted with an empirical baseline from the null permutation analysis. Values in parentheses are units above baseline.

| Brain Representation | L+V | MD+L | MD+V |
| Code Properties | | | |
| --- | --- | --- | --- |
| Dynamic Analysis | 0.22 | 0.32 | 0.31 |
| Static Analysis | 0.19 | 0.21 | 0.20 |

Table 13: Multi-system regression analysis on static and dynamic code properties. Scores represent Pearson correlation between predicted and true code properties.

| Brain Representation | | L+V | MD+L | MD+V |
| Code Models | Empirical Baseline | | | |
| --- | --- | --- | --- | --- |
| Random Embedding | 0.50 | 0.57 (+0.07) | 0.56 (+0.06) | 0.57 (+0.07) |
| Bag Of Words | 0.50 | 0.57 (+0.07) | 0.59 (+0.09) | 0.59 (+0.09) |
| TF-IDF | 0.50 | 0.56 (+0.06) | 0.58 (+0.08) | 0.57 (+0.07) |
| Seq2Seq | 0.50 | 0.55 (+0.05) | 0.57 (+0.07) | 0.56 (+0.06) |
| XLNet | 0.50 | 0.57 (+0.07) | 0.58 (+0.08) | 0.57 (+0.07) |
| CodeTransformer | 0.50 | 0.58 (+0.08) | 0.60 (+0.10) | 0.60 (+0.10) |
| CodeBERTa | 0.50 | 0.60 (+0.10) | 0.62 (+0.12) | 0.62 (+0.12) |

Table 14: Multi-system regression analysis on code model mappings. Scores represent rank accuracy and are contrasted with an empirical baseline from the null permutation analysis. Values in parentheses are units above baseline.

| Code Property | Brain Region A | Brain Region B | t | p | p (corrected) | Is Significant? |
|---|---|---|---|---|---|---|
| Code vs. Sentence | L+V | Language | 3.88 | 3.94e-04 | 5.20e-03 | 1 |
| Code vs. Sentence | L+V | Visual | 3.56 | 9.82e-04 | 7.07e-03 | 1 |
| Code vs. Sentence | MD+L | Language | 3.77 | 5.96e-04 | 5.36e-03 | 1 |
| Code vs. Sentence | MD+L | MD | 3.83 | 4.33e-04 | 5.20e-03 | 1 |
| Code vs. Sentence | MD+V | MD | 3.25 | 2.29e-03 | 1.37e-02 | 1 |
| Code vs. Sentence | MD+V | Visual | 2.96 | 5.36e-03 | 2.54e-02 | 1 |
| Control Flow | L+V | Visual | 2.91 | 5.63e-03 | 2.54e-02 | 1 |
| Control Flow | MD+V | Visual | 4.11 | 1.63e-04 | 5.20e-03 | 1 |
| Data Type | MD+V | Visual | 2.86 | 6.41e-03 | 2.56e-02 | 1 |
| Control Flow | L+V | Language | 0.15 | 8.80e-01 | 1.00e+00 | 0 |
| Control Flow | MD+L | Language | 1.46 | 1.50e-01 | 3.18e-01 | 0 |
| Control Flow | MD+L | MD | -0.11 | 9.16e-01 | 1.00e+00 | 0 |
| Control Flow | MD+V | MD | 0.03 | 9.73e-01 | 1.00e+00 | 0 |
| Data Type | L+V | Language | 1.50 | 1.39e-01 | 3.13e-01 | 0 |
| Data Type | L+V | Visual | 0.44 | 6.59e-01 | 9.49e-01 | 0 |
| Data Type | MD+L | Language | 1.93 | 6.00e-02 | 1.80e-01 | 0 |
| Data Type | MD+L | MD | 0.24 | 8.11e-01 | 9.94e-01 | 0 |
| Data Type | MD+V | MD | 1.76 | 8.67e-02 | 2.23e-01 | 0 |
| Dynamic Analysis | L+V | Language | 0.39 | 7.01e-01 | 9.71e-01 | 0 |
| Dynamic Analysis | L+V | Visual | 1.00 | 3.24e-01 | 6.48e-01 | 0 |
| Dynamic Analysis | MD+L | Language | 1.97 | 5.51e-02 | 1.80e-01 | 0 |
| Dynamic Analysis | MD+L | MD | -0.05 | 9.63e-01 | 1.00e+00 | 0 |
| Dynamic Analysis | MD+V | MD | -0.28 | 7.84e-01 | 9.94e-01 | 0 |
| Dynamic Analysis | MD+V | Visual | 2.51 | 1.56e-02 | 5.61e-02 | 0 |
| Static Analysis | L+V | Language | -0.94 | 3.54e-01 | 6.71e-01 | 0 |
| Static Analysis | L+V | Visual | 1.69 | 9.87e-02 | 2.37e-01 | 0 |
| Static Analysis | MD+L | Language | -0.70 | 4.89e-01 | 8.80e-01 | 0 |
| Static Analysis | MD+L | MD | 0.57 | 5.68e-01 | 9.25e-01 | 0 |
| Static Analysis | MD+V | MD | 0.50 | 6.17e-01 | 9.25e-01 | 0 |
| Static Analysis | MD+V | Visual | 1.76 | 8.50e-02 | 2.23e-01 | 0 |
| Variable Language | L+V | Language | 0.62 | 5.36e-01 | 9.18e-01 | 0 |
| Variable Language | L+V | Visual | -0.00 | 1.00e+00 | 1.00e+00 | 0 |
| Variable Language | MD+L | Language | 0.53 | 5.98e-01 | 9.25e-01 | 0 |
| Variable Language | MD+L | MD | -0.27 | 7.86e-01 | 9.94e-01 | 0 |
| Variable Language | MD+V | MD | -0.22 | 8.28e-01 | 9.94e-01 | 0 |
| Variable Language | MD+V | Visual | -0.00 | 1.00e+00 | 1.00e+00 | 0 |

Table 15: Results from pairwise t-tests of paired brain systems with their individual components for each code property. $+t$ reflects $A > B$, whereas $-t$ reflects $A < B$.

| Code Model | Brain Region A | Brain Region B | t | p | p (corrected) | Is Significant? |
|---|---|---|---|---|---|---|
| Bag Of Words | MD+V | Visual | 3.01 | 4.41e-03 | 2.65e-02 | 1 |
| CodeBERTa | L+V | Language | 2.57 | 1.36e-02 | 4.76e-02 | 1 |
| CodeBERTa | MD+L | Language | 3.40 | 1.45e-03 | 1.52e-02 | 1 |
| CodeBERTa | MD+V | Visual | 2.62 | 1.17e-02 | 4.76e-02 | 1 |
| CodeTransformer | MD+L | Language | 2.84 | 6.80e-03 | 3.51e-02 | 1 |
| CodeTransformer | MD+V | Visual | 3.30 | 1.87e-03 | 1.57e-02 | 1 |
| Seq2Seq | L+V | Visual | 3.02 | 4.21e-03 | 2.65e-02 | 1 |
| Seq2Seq | MD+V | Visual | 3.89 | 3.54e-04 | 9.44e-03 | 1 |
| TF-IDF | L+V | Visual | 2.80 | 7.53e-03 | 3.51e-02 | 1 |
| TF-IDF | MD+L | Language | 3.66 | 6.74e-04 | 9.44e-03 | 1 |
| TF-IDF | MD+V | Visual | 3.72 | 5.49e-04 | 9.44e-03 | 1 |
| XLNet | MD+L | Language | 2.58 | 1.32e-02 | 4.76e-02 | 1 |
| Bag Of Words | L+V | Language | 0.64 | 5.24e-01 | 6.47e-01 | 0 |
| Bag Of Words | L+V | Visual | 1.77 | 8.35e-02 | 1.59e-01 | 0 |
| Bag Of Words | MD+L | Language | 2.25 | 2.92e-02 | 7.65e-02 | 0 |
| Bag Of Words | MD+L | MD | 0.82 | 4.15e-01 | 5.44e-01 | 0 |
| Bag Of Words | MD+V | MD | 0.60 | 5.49e-01 | 6.59e-01 | 0 |
| CodeBERTa | L+V | Visual | 1.46 | 1.52e-01 | 2.45e-01 | 0 |
| CodeBERTa | MD+L | MD | 0.88 | 3.84e-01 | 5.20e-01 | 0 |
| CodeBERTa | MD+V | MD | 1.11 | 2.74e-01 | 4.05e-01 | 0 |
| CodeTransformer | L+V | Language | 1.61 | 1.15e-01 | 1.94e-01 | 0 |
| CodeTransformer | L+V | Visual | 2.18 | 3.48e-02 | 8.47e-02 | 0 |
| CodeTransformer | MD+L | MD | 0.36 | 7.21e-01 | 7.97e-01 | 0 |
| CodeTransformer | MD+V | MD | 0.48 | 6.31e-01 | 7.16e-01 | 0 |
| Random Embedding | L+V | Language | 1.72 | 9.15e-02 | 1.66e-01 | 0 |
| Random Embedding | L+V | Visual | 1.80 | 7.91e-02 | 1.58e-01 | 0 |
| Random Embedding | MD+L | Language | 1.71 | 9.46e-02 | 1.66e-01 | 0 |
| Random Embedding | MD+L | MD | 0.51 | 6.11e-01 | 7.13e-01 | 0 |
| Random Embedding | MD+V | MD | 1.09 | 2.80e-01 | 4.05e-01 | 0 |
| Random Embedding | MD+V | Visual | 2.30 | 2.65e-02 | 7.56e-02 | 0 |
| Seq2Seq | L+V | Language | 1.10 | 2.78e-01 | 4.05e-01 | 0 |
| Seq2Seq | MD+L | Language | 2.43 | 1.99e-02 | 6.41e-02 | 0 |
| Seq2Seq | MD+L | MD | 0.20 | 8.39e-01 | 8.39e-01 | 0 |
| Seq2Seq | MD+V | MD | -0.28 | 7.78e-01 | 8.17e-01 | 0 |
| TF-IDF | L+V | Language | 2.16 | 3.63e-02 | 8.47e-02 | 0 |
| TF-IDF | MD+L | MD | 1.00 | 3.22e-01 | 4.50e-01 | 0 |
| TF-IDF | MD+V | MD | 0.31 | 7.60e-01 | 8.17e-01 | 0 |
| XLNet | L+V | Language | 1.91 | 6.23e-02 | 1.31e-01 | 0 |
| XLNet | L+V | Visual | 2.11 | 4.07e-02 | 8.99e-02 | 0 |
| XLNet | MD+L | MD | 0.79 | 4.35e-01 | 5.53e-01 | 0 |
| XLNet | MD+V | MD | 0.24 | 8.10e-01 | 8.30e-01 | 0 |
| XLNet | MD+V | Visual | 2.29 | 2.70e-02 | 7.56e-02 | 0 |

Table 16: Results from pairwise t-tests of paired brain systems with their individual components for each code model. $+t$ reflects $A > B$, whereas $-t$ reflects $A < B$.

# G    ALL CODE PROPERTIES: SCORES AND CORRELATIONS

We analyzed a series of code properties as part of the static and dynamic code analysis. As several of these properties (*token count*, *node count*, *Halstead difficulty*, *cyclomatic complexity*, and *bytecode ops*) were revealed to be highly correlated in a post-hoc analysis, we report only one measure for this subset in Experiment 1, *token count*, but include all scores here for completeness (Table 17). We also report the correlation matrix between all code properties that led us to select *token count* as the representative property for this subset (Table 18).

| Brain Representation
Code Properties | MD | Language | Visual | Auditory |
|---|---|---|---|---|
| Token Count | 0.17 | 0.24 | 0.09 | 0.00 |
| Node Count | 0.12 | 0.20 | 0.03 | 0.01 |
| Halstead Difficulty | 0.11 | 0.17 | 0.10 | -0.01 |
| Cyclomatic Complexity | 0.18 | 0.24 | 0.10 | 0.01 |
| Bytecode Operations | 0.15 | 0.18 | 0.03 | 0.02 |
| Runtime Steps | 0.33 | 0.20 | 0.17 | 0.12 |

Table 17: Brain region decoding performance on all *static analysis* and *dynamic analysis* code properties. Scores represent Pearson correlation between predicted and true code properties.

| | Datatype | Conditional | Iteration | Tokens | Nodes | Halstead | Cyclomatic | Runtime | Bytecode |
|---|---|---|---|---|---|---|---|---|---|
| Datatype | 1.00 | 0.00 | 0.00 | 0.41 | 0.33 | 0.39 | 0.18 | 0.13 | 0.28 |
| Conditional | - | 1.00 | -0.50 | 0.44 | 0.46 | 0.50 | 0.61 | -0.41 | 0.49 |
| Iteration | - | - | 1.00 | -0.10 | -0.14 | -0.35 | -0.09 | 0.90 | 0.01 |
| Tokens | - | - | - | 1.00 | 0.97 | 0.89 | 0.80 | 0.08 | 0.95 |
| Nodes | - | - | - | - | 1.00 | 0.86 | 0.79 | 0.00 | 0.96 |
| Halstead | - | - | - | - | - | 1.00 | 0.74 | -0.17 | 0.80 |
| Cyclomatic | - | - | - | - | - | - | 1.00 | 0.02 | 0.84 |
| Runtime | - | - | - | - | - | - | - | 1.00 | 0.13 |
| Bytecode | - | - | - | - | - | - | - | - | 1.00 |

Table 18: Correlation matrix across all code properties. The *control flow* property was split into two binary properties here, *conditional* and *iteration*. Of relevance is that *token count* presents with $r > 0.8$ for all *static analysis* properties in the set and as such is used as the representative *static analysis* code property in Experiment 1.

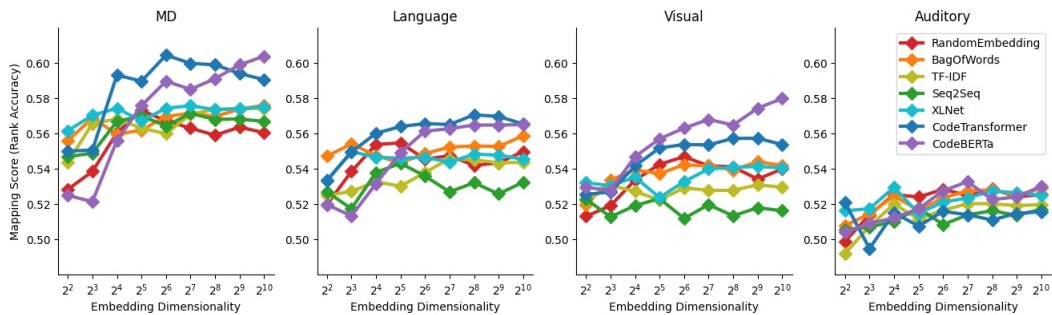

Figure 7: Sensitivity of brain representation mapping to model output dimensions. Each subplot contains the decoding results from a given brain network, and each line reflects a unique code model across a range of controlled embedding dimensions.

## H  SENSITIVITY OF BRAIN MAPPING TO MODEL OUTPUT DIMENSIONS

Of potential interest to the decoding framework is not just the complexity of code models, but their intrinsic dimensionality as well. In order to investigate to what extent brain model mappings are robust to changes in code model dimensionality, and to assess which model representations are most sensitive to compression and expansion, we rerun our current decoding framework while controlling for embedding size.

For each brain network to code model mapping task, prior to the MVPA analysis, we control for the dimensionality of the code model embedding via projection through a Gaussian random matrix $\mathbb{R}^{d_1 \times d_2}$ drawn from $N(0, 1/d_2)$ where $d_1$ is the original code model embedding dimensionality and $d_2$ is the desired dimensionality.

We observe that models of lower complexity (e.g., *bag-of-words*, *TF-IDF*, *seq2seq*, *XLNet*) appear relatively robust to compression, whereas the most complex models (e.g., *CodeTransformer*, *CodeBERTa*) gain considerable performance from higher dimensional expression, and suffer considerably when constrained.

This indicates that higher dimensions of the encoder in complex models encode relevant neural information. Additionally, these effects appear to be most pronounced when decoding from the brain region whose representations yield the strongest mappings, the MD system.

We note here however that we could be observing an interaction effect between model complexity and dimensionality output in these results. Since we cannot fully control for the complexity of these models (by making them all 'equally complex'), this experiment alone cannot drive definitive conclusions.

These results instead constitute a preliminary exploration into the effects of code model dimensionality on brain to model representation mappings, and suggest an avenue for future investigation.

# I ROBUSTNESS OF RESULTS TO REGRESSION METRIC

For the decoding analysis of the continuous valued *dynamic analysis* and *static analysis* properties, it is reasonable to ask why the Pearson correlation metric was chosen as opposed to $RMSE$, as is typically customary for regression tasks. While we present the results using the Pearson correlation metric in the core results for interpretability via the zero-baselne, here we confirm that the use of an $RMSE$ metric leads to the same conclusions. As we see here, the MD, LS, and Visual system decode the *dynamic analysis* property, and MD and LS decode the *static analysis* property with significantly lower $RMSE$ than the null permutation baseline. These results precisely confirm and mirror the patterns observed in Figure 3.

| Brain Network | Code Property | RMSE | Null RMSE | Is Significant? |
|---|---|---|---|---|
| MD | Dynamic Analysis | 3.49 | 4.03 ± 0.08 | 1 |
| MD | Static Analysis | 7.68 | 8.27 ± 0.15 | 1 |
| Language | Dynamic Analysis | 3.68 | 3.95 ± 0.08 | 1 |
| Language | Static Analysis | 7.34 | 8.00 ± 0.15 | 1 |
| Visual | Dynamic Analysis | 3.79 | 4.05 ± 0.08 | 1 |
| Visual | Static Analysis | 7.99 | 8.28 ± 0.16 | 0 |
| Auditory | Dynamic Analysis | 3.79 | 3.98 ± 0.07 | 0 |
| Auditory | Static Analysis | 8.04 | 8.10 ± 0.15 | 0 |

Table 19: $RMSE$ between observed and predicted continuous-valued code properties from the MVPA regression task in Experiment 1, confirming the pattern of results observed in 5.1

