# OpenReview forum: "Representations of Computer Programs in the Human Brain"
_ICLR.cc/2022/Conference — ICLR 2022 Submitted_

### Official Review · Reviewer_mt3z · 2021-11-01

**Correctness:** 3
**Technical Novelty And Significance:** 1
**Empirical Novelty And Significance:** 2
**Recommendation:** 5
**Confidence:** 5

**Main Review:**

Strengths:
- code made publicly available, and the paper also uses publicly available brain data
- mostly clear exposition, with only a few places where more details are needed (see questions below)
- an in-depth and timely empirical investigation

Weaknesses:
- limited methodological novelty
   - this work uses previously established methods for brain decoding, and does not make advances either in the methodology or in the interpretation of existing methodology. By itself, this is not a fatal flaw (a paper can have value to the community without such novelty).
- limited evidence for some of the stated conclusions
   - see Question 1 below for more detailed comments
   - one general comment: when reading the main paper, it is not always clear what differences between brain systems or ML models are actually significant
- limited discussion
  - the investigation of the relationship between language models and the brain recordings has the potential to be very interesting, but the discussion of these results is extremely limited. The paper can benefit from a more thorough discussion of what is learned by the comparison with the language models. For example, it appears that a very simple model like BoW reaches similar decoding performance to that of the most complex model (CodeBERTa)--so then, what do we learn by the comparison with the complex model?

Questions for authors:
1. I have some concerns about the conclusions for 3 of the 4 types of experiments in Section 5.1 under Analysis 1:
   - code vs. sentences: Fig 3 shows that 3 of the brain systems (multi-demand, language, and visual) can significantly distinguish programs according to whether they were written using code or using sentences, with very similar accuracy. In conclusion, the authors state:
> The underlying operations required to solve both types
of programming problems were the same (e.g. both might have
required summing elements in a list). Yet, despite the mental
operations remaining mostly the same, the MD system is able to
discriminate between the two contrasting conditions. This supports the claims made by Liu et al. (2020) and Ivanova et al.
(2020) that the MD system is responsible for code comprehension in addition to being responsive to mentally tracing the execution flow of code snippets–a typical working memory task.

      Since the two conditions are easily distinguishable based on visual properties (as evidence by the very high significant decoding accuracy in the visual system), how can the authors make any claims about what the multi-demand or the language systems are doing based on the decoding accuracy of these two conditions?
    - variable language: Here the authors examine the decoding differences between two conditions, one in which the variable names are in English and the other where the variable names are in Japanese. The authors say that they expect a significant decoding performance in the language system. Would they not expect a significant decoding performance in the visual system? Shouldn't the most obvious difference be in the visual system, because of the systematic visual differences between English and Japanese? Fig 3 shows no significant decoding performance for any brain systems (neither the language nor the visual system). Could this be due to the way the brain data is aggregated over TRs for this analysis?
   - control flow and data type: could the differences in decoding performance for the multi-demand and language systems be due to a difference in the signal-to-noise ratio in the two systems, and how to the authors control for this possible SNR difference between brain systems?
2. Can the authors comment on why they chose to do decoding analyses over encoding analyses (i.e. predicting brain activity from representations of programs)? It seems that encoding could allow for a more in-depth analysis of the differences in what properties different regions/brain systems respond to (i.e. by comparing the weights of the learned encoding models for different brain regions/systems).
3. Can the authors comment on why only left-lateralized regions of interest were examined? Several works in natural language comprehension with naturalistic stimuli show that both the left and right hemispheres are engaged during language comprehension (Wehbe et al. 2014, Huth et al. 2016, Jain and Huth 2018,..) Do the authors expect that the mirror brain regions in the right hemisphere do not engage in the same way during programming comprehension? And if so, do the authors believe that this is a function of their short stimuli (i.e. if the stimuli were longer pieces of code, we would expect both hemispheres to be engaged).
4. One of the questions that was thoughtfully examined by the authors was how the complexity of the ML model affects its relationship with the brain recordings. The authors clarify that what they mean by complexity is the number of model parameters. Another related model property that may affect the relationship with the brain recordings is the dimensionality of the output embeddings--did the authors also examine this property (it's somewhat related to the number of parameters in the model but is not a perfect predictor, i.e. BERT and GPT-2 have the same embedding size but different numbers of parameters overall)?
4. Clarifications needed about:
   - representations of programs extracted from the models: how exactly were they extracted? Was the whole program provided to the model at once? The authors say that the output of the encoder was used, but what about for auto-regressive models (i.e. the last state, or something else?)?
   - brain experiment: was the program presented at once, or word-by-word? for how long? what were the subjects instructed to do?
   - brain data: assuming that each program was presented for multiple TRs, how was the data from multiple TRs handled in the decoding procedure?


**Summary Of The Paper:**

This work examines the relationship between fMRI recordings of people who read short programs and different properties and representations of the programming code. The aim of the work is to understand what properties of code are encoded by different brain systems, and to understand how similar the representations of code in the brain are to those encoded by self-supervised language models that are pretrained to encode programming code. The authors find that several program properties can be significantly decoded from 3 brain systems (the multiple demand system, the language system, and the visual system). They further find that representations of the programs extracted from several machine learning models of varying complexity can also be significantly decoded from these brain systems.




**Summary Of The Review:**

Overall, the paper investigates an interesting and timely question, but offers limited technical and empirical insights. The work can be strengthened by a more in-depth discussion about what is learned by the comparison with complex ML models, and by additional analyses that strengthen the empirical conclusions (see under Questions).

---

> ### Author Response · Authors · 2021-11-17
> **Response to Reviewer mt3z - Part 4**
>
>
> > did the authors also examine this property (it's somewhat related to the number of parameters in the model but is not a perfect predictor, i.e. BERT and GPT-2 have the same embedding size but different numbers of parameters overall)?
>
> Great suggestion!
> We did run an additional set of experiments where we modified the output dimensions of the models.
> **Please see Appendix H of the revised draft we have uploaded for a plot showing the sensitivity of the brain-code representation mapping to output dimensions of these models.**
>
> We found that models of lower complexity (e.g. bag-of-words, TF-IDF, seq2seq, XL-Net) appear relatively robust to compression, whereas the complex models (e.g. CodeTransformer, CodeBERTa) gain considerable performance from higher dimensional expression, and suffer considerably when constrained.
>
> This indicates that higher dimensions of the encoder in complex models encode relevant neural information.
> Additionally, these effects appear to be most pronounced when decoding from the brain region whose representations yield the strongest mappings, the MD system.
>
> We note here however that we could be observing an interaction effect between model complexity and dimensionality output in these results.
> Since we cannot fully control for model complexity of these models (by making them all 'equally complex'), this experiment alone will not yield insightful conclusions.
>
>
> ***
> > representations of programs extracted from the models: how exactly were they extracted? Was the whole program provided to the model at once?
>
> The deep learning architectures we explore in this process text token by token.
> We tokenize our programs using Python's tokenizer utility, which helps remove whitespaces and other non-essential program constructs.
>
> Most models released by the different authors are trained on a common, large, publicly available Python dataset.
> We used these models off the shelf, without any fine-tuning, with the exception of seq2seq, which we trained from scratch ourselves.
>
> We use the raw outputs (logits) from the encoders of each of these models as inputs to our probe models.
>
> The model architectures in turn use the logits from encoders to decode the input (or parts of it) in each of the autoencoder, autoregression, and masked language model settings.
>
> **Proposed draft edit 4 :** We provide details in Appendix B, which we will clarify and update.
>
> ***
> > brain experiment: was the program presented at once, or word-by-word?
>
> The programs were presented at once.
> Each participant had to read through the entire program and had to press a button when they thought they had computed the expected output of the program.
> On the button press, they were presented with four options to choose the correct answer from.
>
> **Proposed draft edit 5 :** We will clarify this in the `Brain and Model Representations` section (section 4) of the draft.
>
> ***
> > brain data: assuming that each program was presented for multiple TRs, how was the data from multiple TRs handled in the decoding procedure?
>
> Following directly from Ivanova et al., data is aggregated across multiple TRs to estimate individual program effects in the following way --
> ```
> ...using a General Linear Model (GLM) in which each experimental condition was modeled with a boxcar function convolved with the canonical hemodynamic response function (HRF). For the localizer experiments, we modeled the entire blocks. For the Python program comprehension experiment, we modeled the period from the onset of the code/sentence problem and until the button press.
> ```
>
> These aggregated individual program activation maps are the basis for our downstream analyses.
>
> **Proposed draft edit 6 :** We will clarify this detail in Appendix B
>
> ***
>
> > Overall, the paper investigates an interesting and timely question, but offers limited technical and empirical insights.
>
> We hope the clarifications we provide above, and the additional discussion we present in our responses above (which will be updated in our draft as well) helps emphasize the merits of our current work.
>
> By the end of the discussion phase, we will have the **6 proposed edits** to our draft.
>
> Until then, we look forward to engaging further and answering any questions or clarifications you may have for us.

---

> > ### Author Response · Authors · 2021-11-23
> > **Draft updated**
> >
> > Dear reviewer
> >
> > We have uploaded an updated draft which has the 5 of the 6 edits we proposed to make. We are still in the process of setting up the encoding analysis--we will keep you posted on the results we see.
> >
> > The edited text has been marked in blue to help track changes.
> >
> > We look forward to addressing any more questions you may have, and engaging in a discussion on the revisions we have made/responses to your initial comments.
> >
> > Appreciate the feedback!
> >
> > Authors

---

> ### Author Response · Authors · 2021-11-17
> **Response to Reviewer mt3z - Part 3**
>
>
> We address below other clarification questions and comments that you had -
>
> > Code vs sentences: Since the two conditions are easily distinguishable based on visual properties (as evidence by the very high significant decoding accuracy in the visual system), how can the authors make any claims about what the multi-demand or the language systems are doing based on the decoding accuracy of these two conditions?
>
> This is a valid observation.
>
> There are two details here which seem conflated given how we have presented this discussion. They are -
> 1. We claim that we have evidence for the MD system being able to decode code vs. sentences, which supports Ivanova et al.'s findings.
> This claim is currently misplaced in the section you have referenced above.
> Rather, the fact that we see `dynamic analysis` is decoded better by the MD as compared to LS (Fig 3, Table 8) supports the observation that the MD system is involved in cognitive processes related to code execution and processing as well.
>
> 2. Given the observation that we see all three - MD, LS, and Vis to significantly decode code vs. sentences with a high accuracy, we ran an additional ablation/sensitivity analysis, which is documented in Appendix F of the submitted draft.
> Table 15, Appendix F shows that combining MD representations with those from the other two systems LS and Vis improves classification of code vs. sentences, suggesting that they represent at least partially orthogonal information.
>
> **Proposed draft edit 2 :** We will clarify this detail, and edit the paragraph you have pointed to.
> Thank you for catching this discrepancy.
>
>
> ***
>
> > Variable language -- shouldn't the most obvious difference be in the visual system, because of the systematic visual differences between English and Japanese?
>
> Not really.
> The Japanese characters were still written out using the English alphabet and not in Kanji, and were placed at arbitrary locations of the program.
> It is very unlikely that these minor changes would induce a noticeable difference in the visual system.
>
> ***
>
> > control flow and data type: could the differences in decoding performance for the multi-demand and language systems be due to a difference in the signal-to-noise ratio in the two systems
>
> Since there exist conditions which can elicit LS > MD, and others which can elicit MD > LS, it is unlikely there exist strong baseline differences in activations between these systems.
>
> Further, if there exist group differences between univariate signals of MD and LS activations (MD > LS, seen in Liu and Ivanova et al.), then its likely to observe the same trend when we decode from these neural activations (our work).
> However, such trends need not necessarily always translate to decoding experiments.
>
> ***
>
> > Can the authors comment on why they chose to do decoding analyses over encoding analyses (i.e. predicting brain activity from representations of programs)?
>
> This is an excellent observation that we do not currently discuss in the draft.
>
> In both our main experiments, our primary goal is to investigate whether specific code properties or representations are encoded in the different brain systems.
>
> An encoding analysis emphasizes understanding the effect of a property or code model representation on an individual voxel in a brain system (interpreted using the learned weights).
>
> While this is an interesting line of analysis, it is not what we wish to address in this work.
> We instead are interested in understanding whether brain systems, modeled as a multivariate representation, encodes code properties or representations learned by code models, irrespective of where they are encoded anatomically.
>
> We however agree that such an encoding analysis will provide complementary information and will improve our current analysis.
>
> **Proposed draft edit 3 :** We are currently attempting this analysis.
> We will update our draft with the results from this analysis if we have them ready by the end of this discussion phase.
>
> ***
>
> > Can the authors comment on why only left-lateralized regions of interest were examined?
>
> In the context of code comprehension, Ivanova et al., 2020 did run a searchlight algorithm to identify if there were any regions, other than the functional ROIs they identified, which showed any activity to code processing.
> The regions identified by the searchlight analysis were subsumed by the functional ROIs.
> The language ROIs they had localized were all left-lateralized.
> We hence use left-lateralized voxels as well.
>
> We do agree that a whole brain analysis will be comprehensive, but since there was no significant code-specific activity in the right-lateralized language regions, it is unlikely it will yield anything substantially different from the current results we have.
>
> It is unclear how the right lateralized regions will respond to longer snippets of code, and that promises to be potentially interesting future work.
>
> ***

---

> ### Author Response · Authors · 2021-11-17
> **Response to Reviewer mt3z - Part 2**
>
>
> Only in our analyses of comparing different brain regions, in looking for whether any one brain system encodes code properties preferentially (Exp 1) or whether any one code model architecture aligns best to the information stored in any one brain system (Exp 2), do we find some instances of weak evidence.
> Additionally, we still observe that MD decodes representations from 3 models significantly better than LS and 4 models significantly better than Vis.
> These data provide evidence that a collection of code models (albeit not all) map more closely to MD than other regions, which is significant and of relevance.
>
> Further, in Experiment 1, MD and LS decode `Control Flow` better than Vis, MD decodes `Dynamic Analysis` better than Vis, and LS decodes `Static Analysis` better than Vis.
> These results suggest that the MD and LS systems definitely encode code-specific properties which go beyond what can be discerned from the visual layout of the code.
> In Experiment 2, we confirm that these encodings are likely driven by token-level information.
> We provide the results of all statistical comparisons in tables 8-11, Appendix E.
>
> The fact that we do not find subsets of brain regions preferentially encoding any subset of properties or models should not invalidate the key finding that it is possible to decode these non-trivial properties and code representations.
>
> ### 3. Implications of results from Experiment 2
>
> The purpose of setting up this experiment was to investigate to what extent the information encoded by extant code models resemble the information encoded by our brains when comprehending code.
> While Ivanova et al (2020) show MD engagement during computer code comprehension, we here probe for its representational content.
> See e.g. Mur et al, 2009, for a discussion of how multivariate techniques, such as decoding, can complement univariate techniques, such as measuring overall response magnitude.
>
> For instance, had we seen the MD system activations to map well only to complex models like codeBERTa, but had seen the LS to selectively map well only to simple models like bag-of-words, we could have possibly concluded that complex code model architectures end up encoding more than tokens and word-level information which the LS is known to model well.
> Inferring from the other direction, we could also have possibly concluded in such a case that the MD system encodes code properties using a representation which does not entirely rely on program tokens.
>
> We found it surprising that the neural information from MD maps to some complex models like seq2seq and code transformers almost as well as to a combination of random token embeddings.
> One plausible explanation for this is that the information we are attempting to decode from MD captures mostly token-level information as against richer structural information about the programs.
> The program stimuli are simple enough to allow the different properties evaluated in our work (`control flow`, `data types`, etc.) to be discerned from token level information alone (as seen in Experiment 1), which is likely why the random embeddings model is also able to predict these properties very well (see Table 4, Appendix C.2).
>
> However, the results we see from Experiment 2 only allow us to conclude that the correspondence we observe between complex models and brain activations is primarily driven by information from tokens present in a program.
> Neither does this imply that the code models do not encode anything more sophisticated than just token information, nor does it imply that the brain does not encode anything more sophisticated.
> Given the number of observations we have, we can only conclude that tokens do get encoded well in both - brain activations and code models.
> This is a new finding on what aspects of code comprehension get encoded by our brains, which will hopefully lead to more works exploring this specific connection in more detail.
>
> **A note on random token embeddings**
>
> We would like to clarify the significance of the random baseline employed in Experiment 2.
> This baseline is essentially a _token projection model_, a meaningful and a strict control to measure the role of tokens when decoding from neural activity.
>
> We also note that none of the related works in NLP (mentioned in Section 2) use such a meaningful baseline to disambiguate token-level effects when analyzing models of varying complexity.
> Simple as it may seem, we believe that this baseline that we introduce in our work is useful in measuring the effect of just the presence of certain tokens.
>
> **Proposed draft edit 1 :** We will add this to the current discussion in Section 6.
>
> ***

---

> ### Author Response · Authors · 2021-11-17
> **Response to Reviewer mt3z - Part 1**
>
>
>
> Thank you very much for your thoughtful and detailed comments.
> We are glad that you found our work interesting and relevant.
> Many of the comments you provide can in fact be addressed by clarifying our overall framing of the results.
> We have addressed your clarification questions below, and will update the manuscript to provide a clearer context for our findings.
>
> ***
>
> ### A summary of key points of feedback and our responses
> The main point of feedback we gather from your comments is on how we could improve our current discussion of the results from Experiment 2, and clearly convey the technical insights learned from this work.
>
> We agree with your observation.
> In addition to providing details on the significance of our results from Experiment 2, we mention below two other important points of discussion.
> We shall add these to the introduction (section 1) and discussion sections (section 6) of the draft.
>
> We also ran additional experiments to observe if the dimensionality of the output embeddings of the different models had any effect on the mapping between brain and code representations.
> Similar to the complexity results, we only find codeBERTa, and to a lesser extent codeTransformers, to be sensitive to the dimensionality.
>
> Detailed comments follow -
>
> ***
>
> > The paper can benefit from a more thorough discussion of what is learned by the comparison with the language models.
>
> Thank you for this suggestion.
> In addition to providing details on the significance of our results from Experiment 2, we mention below two other important points of discussion we think will help clarify the context of all the results we present in this work.
>
> We will modify the draft to include the following three key points of discussion -
>
> ### 1. How are our results different from results reported by Liu et al., 2020 and Ivanova et al., 2020
>
> Liu et al., 2020 and Ivanova et al., 2020 look for group-difference across individuals.
> They investigate the regions of the brain (MD, LS) in which activations to _all_ comprehended code are measured across individuals (Figure 3 in Liu et al. and Figure 2 in Ivanova et al.).
> A primary contribution of their works is the design of the different experiment conditions to disambiguate code comprehension from natural language comprehension, and their key result shows that brain regions do respond to code selectively after controlling for various language-like confounds which codes may exhibit.
>
> Further, Ivanova et al., 2020 do not find any group differences in the activations in response to `control flow` or `data type` properties across individuals (Figure 4 in their work).
>
> It is quite possible that a group difference in activations seen in, say, the MD regions is driven by general attentional demands of comprehending code, and not information about the specific computations and constructs required to process code.
> We provide evidence that these regions encode specific program-related properties as well.
> The results from Experiment 1 helps us conclude that the neural activations pertaining to different programming-constructs like loops and datatypes are _different_ enough between programs comprehended by an individual that it is possible to decode this information from neural activations alone.
>
> **An investigation of individual-level code properties encoded in neural activations across different brain regions has not been previously investigated with the detail our experiment design affords, and we provide the first evidence for the properties indeed being encoded.**.
>
> ### 2. Significance of reported results
>
> We highlight that in both our experiments, we perform two sets of analyses --
> 1. Main analysis - Whether the code properties (Exp 1) or code models (Exp 2) can be decoded from brain activations.
> 2. Additional analysis - Whether any one brain region preferentially encodes any of the code properties or code models.
>
> **The key research question in Exp 1 is whether brain activations even encode code properties, and the key research question in Exp 2 is whether it is even possible to sufficiently decode code model representations.
> We find statistically robust results for both these key questions**.
> It is this set of important results we would like to share with the ICLR community.

---

> > ### Comment · Reviewer_mt3z · 2021-11-18
> > **Remaining question about decoding language model features**
> >
> > I thank the authors for their response. I have not had time to look through the responses to all of my questions, but I wanted to clarify that their first response does not answer my question about what was learned by the decoding analyses using language models. I understand the benefits of doing encoding analyses, as they can allow us to better understand what properties of the code are processed by certain regions. I also understand how that is different from what was done in previous work. It is not clear to me however what was learned by the additional analyses that utilize features extracted from language models that was not learned by the decoding analyses using the hand selected code properties. Just to be clear, I am not saying that decoding analyses with language model features cannot teach us anything beyond hand selected features; what I am saying is that it is not clear to me that this paper uses them in a way that teaches us something more. If the authors can make their argument succinctly, that would be very much appreciated.

---

> > > ### Author Response · Authors · 2021-11-18
> > > **Response clarifying the results from the decoding analysis of language models**
> > >
> > > > It is not clear to me however what was learned by the additional analyses that utilize features extracted from language models that was not learned by the decoding analyses using the hand selected code properties.
> > >
> > > Appreciate the follow up.
> > >
> > > In responding to your first question, we also added in two other points of general clarification which we hoped would provide more context to the point we made on interpreting results from Experiment 2 (decoding analysis using language models).
> > >
> > > Answering your question --
> > >
> > > We learn three pieces of information from our decoding analysis using language models.
> > > 1. It is possible to map various language models from brain activation data.
> > > 2. The quality of this mapping does not seem to be correlated with the complexity of these models.
> > > 3. The quality of this mapping varies between MD and LS significantly for three models - codeBERTa, seq2seq, and TF-IDF.
> > >
> > > We can infer the following from these results -
> > > 1. In predicting _continuous representations_ of programs, afforded by these different models, we do find significant differences in predictions from the MD when compared to LS.
> > > We did not see such a consistent difference for any of the properties we evaluated in Experiment 1.
> > > This suggests that the brain activation data encodes more than just the code properties we evaluated; these other properties are encoded by the different language models.
> > >
> > > 2. We found it surprising that the neural information from MD and LS maps to some complex models like seq2seq and code transformers almost as well as to a combination of random token embeddings.
> > > One plausible explanation for this is that the information we are attempting to decode from MD and LS captures mostly token-level information, as against richer structural information about the programs.
> > > The program stimuli are simple enough to allow the different properties evaluated in our work (control flow, data types, etc.) to be discerned from token level information alone (as validated in Experiment 1), which is likely why the random embeddings model is also able to predict these properties very well (see Table 4, Appendix C.2).
> > >
> > > **This suggests that the information being decoded from brain activations in these two regions is driven _at the least_ by token-level information in the programs.
> > > We cannot come to this conclusion from the code properties investigated in Experiment 1 alone.**
> > >
> > > Note - neither does this imply that the code models do not encode anything more sophisticated than just token information, nor does it imply that the brain does not encode anything more sophisticated. CodeBERTa alone, for which the response is significantly greater than random embeddings, provides evidence that the MD system encodes more than just token information. This will need more investigation. Given the current data and the number of observations we have, we can only conclude that tokens do get encoded well in both - brain activations and code models. This is a new finding on what aspects of code comprehension get encoded by our brains which can be successfully read out using a linear model.
> > >
> > > We provide more details on this in our response `3. Implications of results from Experiment 2` in our comment below titled `Response to reviewer mt3z - Part 2`.

---

> ### Comment · Reviewer_mt3z · 2021-11-30
> **Thanking the authors for response**
>
> I thank the authors for their detailed response to my questions and for the additional experiments. I am still uncertain of the takeaways from the experiments comparing the brain response to representations from NLP models. I do think that the edits have strengthened the paper, and I will increase my score to match that. However, the changes in response to all reviewers appear significant and can benefit from another round of review before they are published.

---

### Official Review · Reviewer_ELAX · 2021-11-02

**Correctness:** 4
**Technical Novelty And Significance:** 2
**Empirical Novelty And Significance:** 3
**Recommendation:** 5
**Confidence:** 4

**Main Review:**

Strengths:
This paper is very clear and well written. It investigates an interesting question of what kinds of information is decodable from brain representations of computer code. Experiments are well thought out and carefully controlled with a decent set of properties of interest and model representation results are compared with relatively extensive sets of models.

Weakness:
I am not sure whether this paper actually provides a lot of new insights for both audiences from the neuroscience community and machine learning community.

It is not surprising to me that the Multiple Demand system in the brain is able to provide above chance decoding performance of the selected properties of the computer code. The Multiple Demand system includes a large part of the prefrontal cortex that is generally responsible for any executive control and cognitive processing human do. It would be more interesting, for example, to run a searchlight algorithm over the brain to pin down specific locations where different properties of computer code are represented.

For experiment 2, other than model complexity, another useful control would be the information integration window (or context window) of different models. The models in comparison have very different mechanism to integrate information across the time during which a computer program is presented. Therefore it is unclear to me how the integration of information over time or length of program is going to affect the brain mapping to these representations. The mapping from brain representations to context window is also potentially helpful for machine learning researchers in designing better models for code representations. However, the analysis in the paper so far, largely limited to decoding based methods and only compared with limited number of controls, are not very informative.



**Summary Of The Paper:**

This paper investigates encoding of computer code in the human brain. The author build decoding models for fMRI responses to predict 1) various properties of python code and, 2) representations of python code derived from different machine learning models. The main conclusion is that the responses from the Multiple Demand system in the brain is capable of provide significance decoding performance of properties and model representations of computer code, such as runtime information.

**Summary Of The Review:**

The paper has done thoughtful analysis and comparisons to investigate how the brain represent computer code and has great potential to be accepted but more in-depth analysis are needed to make it a more informative paper for both the neuroscience and/or machine learning audience.

---

> ### Author Response · Authors · 2021-11-17
> **Response to Reviewer ELAX - Part 2**
>
>
> > It would be more interesting, for example, to run a searchlight algorithm over the brain to pin down specific locations where different properties of computer code are represented.
>
> A searchlight algorithm to identify specific regions of the brain involved in code comprehension and associated cognitive processes emphasizes the anatomical bases of code comprehension.
> This is not the goal of our work, and we instead focus on understanding the nature of code properties that are encoded in the brain (irrespective of where they are encoded anatomically), and whether code model representations mimic the knowledge we encode in our brains.
>
> That said, the brain regions that we do probe for code properties and code model representations are those that have been functionally localized and confirmed using a searchlight-like algorithm by Ivanova et al, 2020.
> So we do probe for code properties and code model representations within the regions which specifically respond to code comprehension.
>
> ***
>
> > For experiment 2, other than model complexity, another useful control would be the information integration window (or context window) of different models. The models in comparison have very different mechanism to integrate information across the time during which a computer program is presented...However, the analysis in the paper so far, largely limited to decoding based methods and only compared with limited number of controls, are not very informative.
>
> That's a thoughtful suggestion.
> We agree that analyzing the context window would be an interesting study.
> However, setting up such an experiment is non-trivial.
> We need to be able to anchor different code contexts to what the individuals visually attended to when presented with the programming task.
>
> There exists no dataset that records both the neural activations in an MRI machine and eye-movements that track what context of the presented stimulus is being attended to, and setting up such a neuroimaging study seems challenging.
> While this is potentially an interesting direction of research, it is well beyond the scope of the work we present here.
>
> **Proposed draft edit 2 :** We will add this discussion in Section 6 of the draft.
>
> ***
> > The paper has done thoughtful analysis and comparisons to investigate how the brain represent computer code and has great potential to be accepted but more in-depth analysis are needed to make it a more informative paper for both the neuroscience and/or machine learning audience.
>
> We urge the reviewer to reconsider the merit of the different findings we present in this work.
> Specifically,
> - we show in Experiment 1 that the MD and LS encode non-trivial code properties like `control flow` and `data type` information of programs.
> We further show that these results are not a consequence of low-level code features like program length.
> No study previously has presented such detailed evidence for neural activations to code comprehension encoding nuanced details of the code being comprehended.
> - we learn three pieces of information from Experiment 2.
> 1. It is possible to map various language models from brain activation data.
> 2. The quality of this mapping does not seem to be correlated with the complexity of these models.
> 3. The quality of this mapping varies between MD and LS significantly for three models - codeBERTa, seq2seq, and TF-IDF.
>
> We infer the following from these results -
> 1. In predicting _continuous representations_ of programs, afforded by these different models, we do find significant differences in predictions from the MD when compared to LS.
> We did not see such a consistent difference for any of the properties we evaluated in Experiment 1.
> This suggests that the brain activation data encodes more than just the code properties we evaluated; these other properties are encoded by the different language models.
>
> 2. The fact that neural information from MD and LS maps to some complex models like seq2seq and code transformers almost as well as to a combination of random token embeddings **suggests that the information being decoded from brain activations in these two regions is driven _at the least_ by token-level information in the programs.
> We cannot come to this conclusion from the code properties investigated in Experiment 1 alone.**
>
> These new findings on the aspects of code comprehension that get encoded by our brains will hopefully help both the neuroscience and ML communities to investigate further the different threads of inquiry and open questions we present in this work.
>
> We will add the **two proposed edits** to the draft by the end of this discussion period.
>
> Until then, we look forward to engaging further and answering any questions or clarifications you may have for us.

---

> > ### Author Response · Authors · 2021-11-23
> > **Draft updated**
> >
> > Dear reviewer
> >
> > We have uploaded an updated draft with the changes you proposed.
> > The updated introduction and discussion section address some of the concerns you shared in your comments.
> > The edited text has been marked in blue to help track changes.
> >
> > We didn't get to engage with you during this discussion phase. We look forward to hearing from you!
> > We are happy to address any more questions you may have, and engage in a discussion on the revisions we have made/responses to your initial comments.
> >
> > Appreciate the feedback!
> >
> > Authors

---

> > ### Comment · Reviewer_ELAX · 2021-11-30
> > **response to authors**
> >
> > Thank you for the response.
> >
> > Part 1 of the response has some merits and thank you for pointing out further differences between this paper and the prior works.
> >
> > For part 2, I do think the context integration window is an important feature to be taken into account. I understand collecting new fMRI data can be beyond reach but manipulating context window parameters in language models and see how that could affect the decoding could be a good start.

---

> ### Author Response · Authors · 2021-11-17
> **Response to Reviewer ELAX - Part 1**
>
>
>
> Thank you for your very perceptive comments.
> We are glad that you found our work interesting and relevant.
>
> Many of the comments you provide can in fact be addressed by clarifying our overall framing of the results.
> We address the key points below and will update the manuscript to provide a clearer context for our findings.
>
> ### A summary of key points of feedback and our responses
> The main point of feedback we gather from your comments is on how we could improve our current discussion of the results from both our experiments, and clearly conveying the technical insights learned from this work.
>
> We agree with your observation.
> We shall provide context to why these results are relevant, and flesh out the discussion on the results we see from the two experiments.
> We shall add these to the introduction (section 1) and discussion sections (section 6) of the draft.
>
> Detailed comments to your questions and clarifications follow --
> ***
>
> > It is not surprising to me that the Multiple Demand system in the brain is able to provide above chance decoding performance of the selected properties of the computer code. The Multiple Demand system includes a large part of the prefrontal cortex that is generally responsible for any executive control and cognitive processing humans do.
>
> To appreciate why it is non-trivial to decode code specific properties from the MD, we first contrast the works of Liu et al., 2020  and Ivanova et al., 2020 with our work.
>
> Liu et al. and Ivanova et al. look for group-difference across individuals.
> They investigate the regions of the brain (MD, LS) in which activations to _all_ comprehended code are measured across individuals (Figure 3 in Liu et al. and Figure 2 in Ivanova et al.).
>
> Interestingly, Ivanova et al., 2020 **do not find any group differences in the activations in response to `control flow` or `data type` properties across individuals (Figure 4 in their work).**
>
> It is quite possible that a group difference in activations seen in, say, the MD regions is driven simply by general attentional demands of comprehending code, owing to the MD system's involvement in general control and cognitive processing as you suggest.
> However, we provide evidence for these regions to encode specific program-related properties as well.
> We show that the code properties we investigate are unique to programs (the presence of `control-flow` constructs, `datatype` constructs, etc.), and are not correlated to other possible low-level code features like the length of a program (Tables 16, 17, Appendix G).
>
> Further, in Experiment 1, we show that MD and LS decode `Control Flow` better than Vis, MD decodes `Dynamic analysis` better than Vis, and LS decodes `Static analysis` better than Vis.
> These results suggest that the MD and LS systems definitely encode code-specific properties which go beyond what can be discerned from the visual layout of the code.
> In Experiment 2, we confirm that these encodings are likely driven by token-level information when we find that the neural activations map almost equally well to simple token-based models and more complex models of code.
>
> **An investigation of individual-level code properties encoded in neural activations across different brain regions has not been previously investigated with the detail our experiment design affords, and we provide the first evidence for the properties indeed being encoded.**
>
> **Proposed draft edit 1 :** We will ensure to highlight this difference in the introduction and discussion sections in our current draft.
>
> ***

---

### Official Review · Reviewer_txju · 2021-11-03

**Correctness:** 3
**Technical Novelty And Significance:** 2
**Empirical Novelty And Significance:** 2
**Recommendation:** 5
**Confidence:** 3

**Details Of Ethics Concerns:**

This research topic is very attractive, beneficial, and important to investigate. Understanding the encoding representation of code properties and semantics of programs in the human brain is crucial to be able to generate artificial models that can reason about code faithfully. This would help to step forward toward bug-free coding and eventually automatic code generation, given the higher-level description of what we need. However, ultimately, there could be some potentially dangerous applications in these areas where you could decode the encoded representations of one's brain images in a reverse procedure and take another step toward intruding human's privacy -- by learning what one is reasoning about in their mind via their brain signals and images.

**Main Review:**

Major comments and questions:
- The results of experiment 1 show the accuracy of the brain representation of code properties. It shows some correlation between brain regions and code properties. However, the t-test results show no significant differences for any brain regions having any preferences for a specific code property. This experiment has been done only on one dataset with Python code. How much is this kind of correlation reliable and pervasive in other datasets?
- As we know, Python is a high-level scripting language that might not be quite distinguishable from short sentences. Have you tried similar experiments on any other datasets or other lower-level languages such as C/C++ or even assembly?
- Is there any improvements in the brain code representations in this work compared to previous work such as Ivanova's given that you are selecting a vector of roughly 1000 voxels for each region?
- Ivanova's paper reported some activities in the MD system during code comprehension. They showed moderate activity in Language systems corresponding to Python code comprehension and no activity regarding visual programming like ScratchJr. They also indicated that LS is responsive to Python code, while visual areas are responsive to ScratchJr. Given that, one other natural experiment to support the mapping brain representation to the code model could be to try it on the brain representations of visual programs (mostly in vision and MD systems) and the corresponding machine-learned encodings using visual recognition models such as convolutional neural nets. Have you tried such experiments on ScratchJr codes similar to what you have done for Python codes?

Minor comments:
- The explanation in the paper is fine, however, adding more diagrams of the whole procedure of the framework as well as mathematical descriptions of the models and their input and output as matrices/vectors would have helped a lot to understand the whole idea faster and easier.

**Summary Of The Paper:**

This paper introduces a systematical framework to discover the relationship between the brain representations of programs and their corresponding code models. This framework helps us to understand the code properties encoded in the human brain so that we could evaluate whether ML models faithfully represent human brain representations of computer code comperhensions.
This paper focuses on answering two questions by showing the results of related experiments on a dataset of 72 programs and 24 persons' brain recordings: First of all, the authors show that how well each of the four brain systems considered in this paper including Multiple Demand, Language, Vision, and Auditory systems encode specific code properties using a ridge regressor. Then they demonstrate another ridge regressor that can map brain representations to the corresponding learned representations by computational language models of code with different model complexity.


**Summary Of The Review:**

The paper is well written, the motivations are clear, and the references are enough. Related work is adequately established the existing research in the field and compared the goal of the current research to previous work. However, the ideas of the paper are incremental and mostly applications of existing methods which have been put together to approach an interesting open problem. Moreover, usually further experiments on more datasets are required to evaluate any correlation between the code models and the brain representations in the proposed framework.

---

> ### Author Response · Authors · 2021-11-17
> **Response to reviewer txju - Part 3**
>
>
> > One other natural experiment to support the mapping brain representation to the code model could be to try it on the brain representations of visual programs (mostly in vision and MD systems) and the corresponding machine-learned encodings using visual recognition models such as convolutional neural nets. Have you tried such experiments on ScratchJr codes similar to what you have done for Python codes?
>
> This is a great suggestion!
> We intend to extend our investigation to programming languages with fully visual interfaces.
> However, this merits a separate study because the two experiments we formulate in this work cannot be naturally extended to the current ScratchJr. dataset by Ivanova et al. since the programs there are too simple for any meaningful analysis of the performance of code models.
> In work ahead, we plan to study what code properties are encoded by activations in response to such languages.
>
> >  However, ultimately, there could be some potentially dangerous applications in these areas where you could decode the encoded representations of one's brain images in a reverse procedure and take another step toward intruding human's privacy -- by learning what one is reasoning about in their mind via their brain signals and images.
>
> Great point! We agree that this line of work poses a potential threat to privacy and offers an important ethical question to address.
>
> On a related note, with neuroimaging-based probing methods though, it has been shown that it is relatively easy to always deceive such detectors from detecting any real mental states.
> See Miller, Greg. "fMRI lie detection fails a legal test." (2010): 1336-1337 for details and relevant references.
>
> **Proposed draft edit 3 :** Space permitting, we will include this discussion in Section 6 of the draft.
> ***
>  > The explanation in the paper is fine, however, adding more diagrams of the whole procedure of the framework as well as mathematical descriptions of the models and their input and output as matrices/vectors would have helped a lot to understand the whole idea faster and easier.
>
> Thanks for the suggestion. We'll work on this.
>
> **Proposed draft edit 4 :** We will attempt to annotate the main figure better and add more details as you suggest.
>
> ***
>
> In light of these responses, we urge the reviewer to reconsider the merit of the different findings we present in this work.
> Specifically,
> - we show in Experiment 1 that the MD and LS encode non-trivial code properties like `control flow` and `data type` information of programs.
> We further show that these results are not a consequence of low-level code features like program length.
> No study previously has presented such detailed evidence for neural activations to code comprehension encoding nuanced details of the code being comprehended.
> - we learn three pieces of information from Experiment 2.
> 1. It is possible to map various language models from brain activation data.
> 2. The quality of this mapping does not seem to be correlated with the complexity of these models.
> 3. The quality of this mapping varies between MD and LS significantly for three models - codeBERTa, seq2seq, and TF-IDF.
>
> We infer the following from these results -
> 1. In predicting _continuous representations_ of programs, afforded by these different models, we do find significant differences in predictions from the MD when compared to LS.
> We did not see such a consistent difference for any of the properties we evaluated in Experiment 1.
> This suggests that the brain activation data encodes more than just the code properties we evaluated; these other properties are encoded by the different language models.
>
> 2. The fact that neural information from MD and LS maps to some complex models like seq2seq and code transformers almost as well as to a combination of random token embeddings **suggests that the information being decoded from brain activations in these two regions is driven _at the least_ by token-level information in the programs.
> We cannot come to this conclusion from the code properties investigated in Experiment 1 alone.**
>
> These new findings on the aspects of code comprehension that get encoded by our brains will hopefully help both the neuroscience and ML communities to investigate further the different threads of inquiry and open questions we present in this work.
>
> We will modify the draft and include the **4 proposed edits** discussed above.
>
> Until then, we look forward to engaging further and answering any questions or clarifications you may have for us.

---

> > ### Author Response · Authors · 2021-11-23
> > **Draft updated**
> >
> > Dear reviewer
> >
> > We have uploaded an updated draft with the changes you proposed.
> > The updated discussion section addresses some of the concerns you shared in your comments.
> > The edited text has been marked in blue to help track changes.
> >
> > We didn't get to engage with you during this discussion phase. We look forward to hearing from you!
> > We are happy to address any more questions you may have, and engage in a discussion on the revisions we have made/responses to your initial comments.
> >
> > Appreciate the feedback!
> >
> > Authors

---

> ### Author Response · Authors · 2021-11-17
> **Response to reviewer txju - Part 2**
>
>
> We also add that none of the related works in NLP (mentioned in Section 2) use such a meaningful baseline to disambiguate token-level effects when analyzing models of varying complexity.
> Simple as it may seem, we believe that this baseline that we introduce in our work is useful in measuring the effect of just the presence of certain tokens.
>
> **Proposed draft edit 1 :** We will include this discussion in Section 6 of the draft.
>
> ***
>
> > This experiment has been done only on one dataset with Python code. How much is this kind of correlation reliable and pervasive in other datasets?
>
> We do not have enough studies done in multiple languages, and this is an open problem for the neuroimaging community to contribute towards.
> A particular drawback of neuroimaging experiments is its steep financial and time demands.
> Since we are the first few to investigate such questions and establish these experiments, we are limited by the resources available to us.
>
> Despite this, we do analyze well curated, well studied datasets of brain activity and find statistically significant results for both the key questions we frame and investigate in this work.
>
> Additionally, the public release of our analysis pipeline should enable other researchers to evaluate their datasets against our benchmark of code properties and code models, which will help provide comprehensive answers to the questions we investigate in this work.
>
> **Proposed draft edit 2 :** We will include this discussion in Section 6 of the draft.
>
> ***
>
> > As we know, Python is a high-level scripting language that might not be quite distinguishable from short sentences. Have you tried similar experiments on any other datasets or other lower-level languages such as C/C++ or even assembly?
>
> > Is there any improvements in the brain code representations in this work compared to previous work such as Ivanova's given that you are selecting a vector of roughly 1000 voxels for each region?
>
> We address both these questions here.
> It is first worth appreciating the difference in the general design of our work and that of Liu et al., 2020 and Ivanova et al., 2020.
> That will help provide context to the specific responses to these two questions.
>
> Liu et al. and Ivanova et al. look for group-difference across individuals.
> They investigate the regions of the brain (MD, LS) in which activations to _all_ comprehended code are measured across individuals (Figure 3 in Liu et al. and Figure 2 in Ivanova et al.).
> A primary contribution of their works is the design of the different experiment conditions to disambiguate code comprehension from natural language comprehension.
>
> Hence, Both Liu et al., 2020 and Ivanova et al., 2020 conclusively show through their thorough experiment design that neural responses to Python are significantly different from neural responses to natural language even after controlling
> for various "text-like" properties of the programs.
>
> Our work investigates a different set of questions.
> We are interested in understanding whether the neural activations observed in response to code comprehension are driven by non-trivial code properties, and whether code models mimic what our brains encode.
> In fact, in their group-level analysis, Ivanova et al. do not find any group differences in the activations in response to `control flow` or `data type`, properties that we investigate in our work, across individuals (Figure 4 in their work).
>
> In light of these differences --
> 1. Your concern that neural activations to Python could be confounded by language-like properties of Python has been comprehensively addressed in the experiment designs of Liu et al. and Ivanova et al.
> However, it is a great suggestion for the neuroimaging community at large to establish the generalizability of their results across different programming languages.
>
> 2. We sample from nearly 1000 most relevant brain voxels in each individual subject to test whether neural activations in the different brain systems encode either code properties or code model representations.
> This sampling is done from the nearly 50,000 (top 10%) voxels which responded to different control conditions in different brain systems like MD and LS (details in Ivanova et al., 2020).
> We perform this sampling to reduce the number of independent variables which we feed into our linear probe models, and make it comparable to the number of program samples for each subject (48 in total).
> This downsampling does not affect the performance of the linear probes nor significantly changes the quality of the program representation in the different brain systems; we had internally tried multiple experiments by considering the full 50,000 voxel set and had seen comparable results.
> ***

---

> ### Author Response · Authors · 2021-11-17
> **Response to reviewer txju - Part 1**
>
>
>
> Thank you for your helpful comments.
> We are glad that you found our work interesting and relevant.
> Many of the comments you provide can in fact be addressed by clarifying our overall framing of the results.
> We address the key points below and will update the manuscript to provide a clearer context for our findings.
>
> ### A summary of key points of feedback and our responses
> The key point of feedback we gather from your comments is that it is unclear how to interpret some of the results in both Experiments 1 and 2, given how some comparisons reported are not statistically significant.
> Further, it may not be clear how our experiment complements the findings of Liu et al., 2020 and Ivanova et al., 2020.
>
> We respond to these important questions below in detail, in addition to addressing your other thoughtful comments.
>
> A summary of our response to your key concern is -- in both Experiment 1 and 2, we clarify that some tests of _differences_ in key experiment conditions end up being insignificant.
> The key research questions we wish to answer in this work do not pertain to studying these differences, but rather asserting the presence of specific code-related properties in brain representations of code.
> All the key results in Experiment 1 are statistically significant.
> In Experiment 2, when specifically comparing models of different complexity (one of the key questions we want to answer), we agree that we do not have statistically significant results to back the general trends we see.
>
> Detailed comments are mentioned below -
>
> ***
>
> > The results of experiment 1 show the accuracy of the brain representation of code properties. It shows some correlation between brain regions and code properties. However, the t-test results show no significant differences for any brain regions having any preferences for a specific code property.
>
> This is an important observation.
> We admit that we have not dealt with clarifying this important detail in our discussion section, and propose to modify it during this discussion phase.
>
> We first highlight that in both our experiments, we perform two sets of analyses --
> 1. Main analysis - Whether the code properties (Exp 1) or code models (Exp 2) can be decoded from brain activations.
> 2. Additional analysis - Whether any one brain region preferentially encodes any of the code properties or code models.
>
> **The key research question in Exp 1 is whether brain activations even encode code properties, and the key research question in Exp 2 is whether it is even possible to sufficiently decode code model representations.
> We find statistically significant results for both these key questions.**
> It is this set of important results we would like to share with the ICLR community.
>
> Only in our analyses of comparing different brain regions, in looking for whether any one brain system encodes code properties preferentially (Exp 1) or whether any one code model architecture aligns best to the information stored in any one brain system (Exp 2), do we find weak evidence.
> The fact that we do not find brain regions preferentially encoding any one set of properties should not invalidate the key finding that it is possible to decode these non-trivial properties and code representations. That said, we do find a preferential encoding of model representations (Exp 2), which informs us that the MD system definitely encodes token-based information, a property which was not directly testable using Exp 1.
>
>
> **A note on the random baseline we employ in Experiment 2**
>
> We would like to clarify the significance of the random baseline employed in Experiment 2.
> This baseline is essentially a _token projection model_, a meaningful and a strict control to measure the role of tokens when decoding from neural activity.
> The decoding performance not being significantly different from this model suggests that the brain activity we observe and decode from predominantly just encodes token-level information, and not something structural about the programs which we expect complex model architectures to encode.

---

> ### Comment · Reviewer_txju · 2021-12-09
> **Thanking and responding to authors**
>
> I appreciate the authors for their well-written responses regarding my questions and concerns. They provide convincing arguments about the differences between this paper and previous work. Moreover, they discussed further merit points to interpret some of the results in both Experiments 1 and 2,
> Acknowledging great achievements by authors toward this interesting yet hard research topic, and admiring their significant endeavour in this regard, I am still uncertain about the takeaways of this article without additional investigations to back the current results.

---

### Official Review · Reviewer_vpX5 · 2021-11-03

**Correctness:** 3
**Technical Novelty And Significance:** 3
**Empirical Novelty And Significance:** 2
**Recommendation:** 5
**Confidence:** 4

**Main Review:**

##########

Updated review (apologies for the delay!):

I commend the authors for engaging thoroughly with the reviews, providing new evidence and making convincing arguments. Having gone through their updated draft, rebuttals and new identification experiment (for the visual system confound), I am convinced that the MD/LS results show code processing distinct from pure visual features and that the random baseline performance in result 2 is perhaps an effect of the simple dataset, not that the regions/models only care about token-level information. To this end, I am increasing my score by 2 points.

Having said that, I agree with the other reviewers that this has been a productive discussion and the paper could benefit from more additions- particularly, stronger/cleaner results to show what information is differentially processed in MD vs. LS etc.. I am still unconvinced that the current approach is more promising in deciphering brain function than encoding (with variance partitioning). Hence I am unable to fully endorse this paper at this time.

##########

Strengths: The paper was well-written and the research question is interesting + useful. The use of publicly available data and code release is beneficial to the community.

The authors note that the visual ROI can effectively classify 4 different stimulus properties but it does not significantly predict the token count. This to me suggests that the 4 properties evaluated here are in fact strongly confounded with program length, presence of letters etc. and may not specifically correspond to code comprehension. What do the authors think about such confounds being relevant in other areas as well and its impact on the conclusions reached in the paper?

Overall, I am not convinced that classification or identification accuracy is useful in gauging what properties of code are encoded by different brain regions. It would be useful to introduce more contrasts or control for alternate exploratory variables. While the authors allude to several of the metrics being correlated (like in the static analyses and byte code), not only would it be useful to add a correlation matrix but also use a more systematic approach like variance partitioning to see what unique properties of code these feature spaces explain. (Both against each other and confounds like program length) The authors are making definitive statements about the observed performance patterns based on prior knowledge of the MD system, LS system etc. However, given the possibility of confounds and correlations between the metrics itself, I am unable to view these results as strong evidence for one mechanism over the other.

Re experiment 1:
- Why isn’t the baseline just chance probability instead of a round-about way with randomly sampled assignments? For example, it should 1/3 for control flow or 0.5 for variable language.
- There are no details on how the functional ROIs were identified: did you use specific contrasts, FreeSurfer annotations etc.? If indeed AC refers to primary AC,  perhaps the significance test being used is weak and cannot meaningfully identify signifiant effects.

Re Analysis 1:
1. Ivanova et al., 2020 effectively demonstrate that the multi-demand network uniquely responds to code. However, it is not clear to me what new findings are proposed in the current work beyond this.
2. Control flow and data type: The authors make claims about accuracy differences between MD & LS. However, these differences are not statistically signifiant by their own admission:
> We follow this up by using t-tests to examine whether any one brain region decodes any given property more significantly than another, but find no differences between the MD system and the Language system for any properties (Table 8, Appendix E).
3. Same comment holds for the contrast in dynamic and static analyses:
> We additionally test if any brain region has a preference for a specific code property over another. For instance, is the evidence seen in Figure 4, of MD more accurately decoding dynamic analysis properties than static analysis properties, statistically significant? Likewise, does the LS predict static analysis properties significantly more accurately than the dynamic analysis properties? We do not find any significant differences (Table 9, Appendix E).

Re Experiment 2:
- What layer of the models were used? Is it the logits or softmax output?
- Can the authors provide more details on how rank accuracy was computed and how the regression task was set up?
- It is unclear to me what scientific question this experiment is answering: Yes, different brain regions have good identification accuracy for different brain models, but isn’t this expected from a) prior work showing these regions respond to code and 2) these models capturing non-linear features of the code stimuli? What does this tell us about brain function?
- I found weak evidence for the identification accuracies presented here since several of the tests were not statistically significant and did not beat the random embedding baseline.
- Re the following:
> The fact that MD maps to other complex models other than CodeBERTa, like CodeTransformer and seq2seq, as accurately as the Random embedding model is a strong indicator that the representations in these brain systems is strongly driven by token-level information in the input programs.

I do not follow this claim. Do the authors mean brain systems or code models? Couldn't the same result be observed if the high-level code information was not linearly decodable as permissible by the linear regression setting here?

Other questions:
1. Since the magnitude of the predicted value is important here, I am confused by the use of linear correlation as opposed to RMSE.
2. For discrete properties, this is clearly not regression. The authors should clarify that they build classification/regression models with an L2 penalty.
3. What do the authors mean by “zero-shot” here since iiuc, the model was never actually tested on stimulus decoding following training on code representation identification?
> We show that it is possible to perform zero-shot decoding of the computer program being comprehended using a proxy representation produced by a suite of code models.
4. How were code representations extracted from XLNet and seq-2-seq? Were the models fine-tuned on code?

**Summary Of The Paper:**

In this work, the authors explore the relationship between 4 different brain regions (multi-demand network, language network, visual system, auditory system) and different features of program code. Specifically, they look at hidden state representations of code language models (seq2seq, CodeBERTa, CodeTransformer, XLNet), tf-idf and BoW representations of the input. For non-LM based features, they look at a code vs. sentence contrast, variable language (English vs. Japanese variable names), data types (strings vs. numerals) and control flow (for vs. if vs. no branching) To analyze relationships between the BOLD signal and stimulus features, they build linear classifiers/regressors from the BOLD activity of each system to each of the stimulus features. Models are evaluated using classification accuracy and linear correlation for regression. In the case of hidden-state features, model are evaluated using rank accuracy.

Overall, the authors find that the visual system is capable of significantly predicting several of the hand-crafted code features suggesting that these features are correlated with low-level stimulus properties like program length. While differences between MD & LS are not significant, these models successfully predict 5/6 hand-crafted features. In the code representation prediction task, the authors find that the MD, LS and Visual systems are able to rank significantly above change. However, the LS and visual systems do not beat a random token embedding baseline. Aside from CodeBERTa, the MD system is also not significantly above the random baseline.

**Summary Of The Review:**

While the proposition of understanding what aspects of code processing are represented in different brain regions is interesting, in its current form, I think the paper doesn’t successfully disentangle this. Several of the effects reported are not significant above chance or random baselines and further, not significantly different across brain regions. The features considered are also confounded with low-level properties of code comprehension that haven’t been controlled for. To this end, it is hard to infer what properties different systems are encoding and if there is a take away beyond prior work that identifies regions robustly responding to code.

---

> ### Author Response · Authors · 2021-11-17
> **Response to reviewer vpX5 - Part 7**
>
>
> > For discrete properties, this is clearly not regression. The authors should clarify that they build classification/regression models with an L2 penalty.
>
> We mention the following in Section 4.2 in our draft.
> ```
> The brain representations (Section 4.1) are mapped to each of the code properties by training a ridge regression model each for every participant-property pair. To evaluate model performance, we use classification accuracy when the predicted values are categorical (e.g. string vs. numeric data types), and the Pearson correlation coefficient when the predicted values are continuous (e.g. number of runtime steps).
> ```
>
> **Proposed draft edit 8 :** We will modify this to clarify classification model vs. regression model.
> As mentioned earlier, we will also clarify the use of RMSE vs. Pearson correlation for ordinal-valued predictions.
>
> ***
>
> > What do the authors mean by “zero-shot” here since iiuc, the model was never actually tested on stimulus decoding following training on code representation identification?
> ```We show that it is possible to perform zero-shot decoding of the computer program being comprehended using a proxy representation produced by a suite of code models.```
>
> We do test stimulus-decoding following training on code representation identification--this is the central setup of Experiment 2 (Figure 2).
>
> Our setup comprises the following workflow -
> 1. For each subject, we have brain representations for each of the 48 programming stimuli they saw.
> 2. For each stimulus, we also have model representations predicted by code models.
> 3. Using a subset of the data in step 1., we train a linear probe to predict 2. from 1.
> We test the performance of this probe on the remaining subset of stimuli in 1.
>
> The _zero-shot_ we mention here refers to the fact that the linear probe predicts code representations of unseen programs in the held-out dataset from just the neural representations seen during training.
> This is not technically _zero-shot_ in the traditional sense of predicting unseen labels/categories, but we do show that novel programs can be individually identified on a first try using only brain recordings, which is significant and relevant.
>
> **Proposed draft edit 9 :** We will clarify this detail in the modifications we make during this discussion phase.
>
> ***
>
> > How were code representations extracted from XLNet and seq-2-seq? Were the models fine-tuned on code?
>
> seq2seq was pre-trained on a code corpus.
> The XLNet model released by the authors are trained on natural language questions-code pairs.
> We used the model off the shelf, without any fine-tuning, and extracted the logits from its encoder.
> Details are provided in Appendix B, subsection _Code models - Configuration details._
> See also https://github.com/anonmyous-author/anonymous-code/blob/main/braincode/encoding.py#L231-L253 for details on XL-Net.
>
> **Proposed draft edit 10 :** We will mention these details in the main draft as well.
>
> ***
> ### Summary
> The reviewer's conclusion are the following --
> > Several of the effects reported are not significant above chance or random baselines and further, not significantly different across brain regions. The features considered are also confounded with low-level properties of code comprehension that haven’t been controlled for. To this end, it is hard to infer what properties different systems are encoding and if there is a take away beyond prior work that identifies regions robustly responding to code.
>
> **We hope that our detailed responses above convince you that the claims we make are not confounded by low-level properties of code, are not statistically insignificant, and suggest that the brain does indeed encode non-trivial code properties, while also encoding token-level information which can be mapped to ML-based code models.**
>
> We shall work on the **10 proposed draft edits** we mention above, and will have these changes reflected by the end of this discussion period.
>
> Until then, we look forward to engaging further and answering any questions or clarifications you may have for us.

---

> > ### Comment · Reviewer_vpX5 · 2021-11-20
> > **Zero-shot decoding**
> >
> > This is a nit (apologies) but the experiment doesn't involve actually reconstructing (decoding) the "computer program", just the representation from CodeBERTa etc. and hence my confusion with the wording!

---

> > > ### Author Response · Authors · 2021-11-22
> > > **Clarified**
> > >
> > > Noted. We have clarified this in our draft.

---

> > ### Author Response · Authors · 2021-11-23
> > **Updated draft**
> >
> > Dear reviewer
> >
> > We have uploaded an updated draft which has the 10 edits we proposed to make.
> > The edited text has been marked in blue to help track changes.
> >
> > We look forward to addressing any more questions you may have, and engaging in a discussion on the revisions we have made/responses to your initial comments.
> >
> > Appreciate the feedback!
> >
> > Authors

---

> ### Author Response · Authors · 2021-11-17
> **Response to reviewer vpX5 - Part 6**
>
>
> ### Responses to other minor clarifications
>
> > Why isn’t the baseline just chance probability instead of a round-about way with randomly sampled assignments? For example, it should 1/3 for control flow or 0.5 for variable language.
>
> Thanks for catching that.
> What we report are the empirically established baselines on running the null-permutation tests 1000 times.
> They should, in the limit, converge to 1/3 for control flow, and 1/2 for variable language.
>
> **Proposed draft edit 5 :** We will change the baselines on the plots to reflect 1/3 and 1/2 respectively, which agreeably would be easier to understand.
>
> ***
>
> > There are no details on how the functional ROIs were identified: did you use specific contrasts, FreeSurfer annotations etc.? If indeed AC refers to primary AC, perhaps the significance test being used is weak and cannot meaningfully identify significant effects.
>
> We provide these details in Appendix B.
> We reproduce some of those details here --
> ```
> For each trial in the fMRI experiment, stimulus responses in each voxel were extracted from the parameters of a General Linear Model (GLM) fit to the time-varying BOLD signal. The predictors for the GLM included trial ID (equivalent to problem ID), run number, and motion regressors.
> The voxels were then filtered using gray-matter masking and (for the MD and LS) network localization.
> Although fMRI measurements return whole brain responses, only a thin layer of cortex dubbed gray matter contains BOLD signal of interest to these analyses.
> Gray matter voxels were selected using a Bayesian segmentation of the anatomical brain image into standard tissue types, and then returning the set of indices where the posterior gray matter probability exceeds 0.70 [Ashburner et al., 1999].
> For the visual and auditory networks, primary sensory areas were identified using an anatomical atlas [Rolls et al., 2020].
> For the MD and LS, voxels were functionally localized as those containing the top 10\% of responses to their respective functional localizer tasks, as described in Ivanova et al., 2020.
> See Fedorenko et al., 2010 for a discussion of the functional localization approach as it pertains to the language network.
> GLM modeling and gray matter segmentation was performed using SPM12; functional localization was performed using the toolbox released by Ivanova et al., 2020.
> Once voxel responses within each brain region were extracted for each trial of each run, some additional preprocessing was required before finalizing the brain representations, and passing them to downstream models.
> ```
>
> ***
>
> > What layer of the models were used? Is it the logits or softmax output?
>
> We use the raw outputs (logits) from the encoders of each of these models as inputs to our probe models.
>
> The model architectures in turn use the logits from encoders to decode the input (or parts of it) in each of the autoencoder, autoregression, and masked language model settings.
>
> **Proposed draft edit 6 :** We will add this detail in the draft.
>
> ***
>
> > Can the authors provide more details on how rank accuracy was computed and how the regression task was set up?
>
> We provide these details in Appendix B, section _MVPA cross-validation and hyperparameters_.
> Reproducing the description here --
> ```
> For the model representations, we report rank accuracy between model representation predictions and the true model representations of left-out samples. We calculate rank accuracy as the percentile of a predicted and true target pairing within the full set of possible pairings sorted by a Euclidean distance metric. Rank accuracy was chosen for this experiment as it returns a more interpretable 50% baseline
> through which to evaluate the brain to model mappings. Following metric calculation on each cross-validation fold, we take the mean of those estimates to report an out-of-sample score over the entire dataset for each subject. We then take the mean of those scores across subjects to derive an overall performance measure.
> ```
>
> The rank accuracy score is commonly used in information retrieval in situations where there are several elements in a range that are similar to the correct one (closely related programs in our case).
> In our setting, the score assigns full credit to getting the exact stimulus program at the top of the ranking, and partial credit if the top-ranked program is _close_ to the target.
>
> **Proposed draft edit 7 :** We will add these details in Section 3 of the draft.
>
> ***

---

> ### Author Response · Authors · 2021-11-17
> **Response to reviewer vpX5 - Part 5**
>
> We address here another perceptive question you raised --
> > Do the authors mean brain systems or code models? Couldn't the same result be observed if the high-level code information was not linearly decodable as permissible by the linear regression setting here?
>
> We did mean brain systems.
> The complexity of the probe model can also possibly explain these results.
> While we agree that complex probes may help uncover complex structural information being encoded in the neural activations, the constraints imposed by neuroimaging studies in the amount of brain-activation data we have allows us to only confirm that token-level information can be linearly read out from both the MD and the language systems.
> Successfully training non-linear models would require an order of magnitude more fMRI recordings, acquiring which is an expensive and a time consuming process.
>
> **Proposed draft edit 3 :** We will add this to the current discussion in Section 6.
>
> ***
>
> ### Motivation for Experiment 2
>
> > It is unclear to me what scientific question this experiment is answering: Yes, different brain regions have good identification accuracy for different brain models, but isn’t this expected from a) prior work showing these regions respond to code and 2) these models capturing non-linear features of the code stimuli? What does this tell us about brain function?
>
> The purpose of setting up this experiment was to investigate to what extent the information encoded by extant code models resemble the information encoded by our brains when comprehending code, and what it can tell us about the properties driving this function in our brains.
>
> **What do we learn from this experiment?**
>
> We observe the following three results -
> 1. It is possible to map various language models from brain activation data.
> 2. The quality of this mapping does not seem to be correlated with the complexity of these models.
> 3. The quality of this mapping varies between MD and LS significantly for three models - codeBERTa, seq2seq, and TF-IDF.
> We can infer the following from these results -
>
> We can make the following inferences --
>
> 1. In predicting _continuous representations_ of programs, afforded by these different models, we do find significant differences in predictions from the MD when compared to LS. We did not see such a consistent difference for any of the properties we evaluated in Experiment 1. This suggests that the brain activation data encodes more than just the code properties we evaluated; these other properties are encoded by the different language models.
>
> 2. We found it surprising that the neural information from MD and LS maps to some complex models like seq2seq and code transformers almost as well as to a combination of random token embeddings. One plausible explanation for this is that the information we are attempting to decode from MD and LS captures mostly token-level information, as against richer structural information about the programs. The program stimuli are simple enough to allow the different properties evaluated in our work (control flow, data types, etc.) to be discerned from token level information alone (as validated in Experiment 1), which is likely why the random embeddings model is also able to predict these properties very well (see Table 4, Appendix C.2).
>
> **This suggests that the information being decoded from brain activations in these two regions is largely driven by token-level information in the programs. We cannot come to this conclusion from the code properties investigated in Experiment 1 alone.**
>
> Note - neither does this imply that the code models do not encode anything more sophisticated than just token information, nor does it imply that the brain does not encode anything more sophisticated. CodeBERTa alone, for which the response is significantly greater than random embeddings, provides evidence that the MD system encodes more than just token information. This will need more investigation. Given the current data and the number of observations we have, we can only conclude that tokens do get encoded well in both - brain activations and code models. This is a new finding on what aspects of code comprehension get encoded by our brains which can be successfully read out using a linear model.
>
> **A note on the random baseline we employ in Experiment 2**
>
> We would like to clarify the significance of the random baseline employed in Experiment 2.
> This baseline is essentially a _token projection model_, a meaningful and a strict control to measure the role of tokens when decoding from neural activity.
> None of the related works in NLP (mentioned in Section 2) use such a meaningful baseline to disambiguate token-level effects when analyzing models of varying complexity.
>
> **Proposed draft edit 4 :** This important discussion is currently missing section 6. We will modify the draft to include it.
>
> ***

---

> ### Author Response · Authors · 2021-11-17
> **Response to reviewer vpX5 - Part 4**
>
>
> ### How are our results different from results reported by Liu et al., 2020 and Ivanova et al., 2020
>
> > Ivanova et al., 2020 effectively demonstrate that the multi-demand network uniquely responds to code. However, it is not clear to me what new findings are proposed in the current work beyond this.
>
> Great question!
>
> The differences are along the lines discussed in the previous section.
> Liu et al., 2020 and Ivanova et al., 2020 look for group-difference across individuals.
> They investigate the regions of the brain (MD, LS) in which activations to _all_ comprehended code are measured across individuals (Figure 3 in Liu et al. and Figure 2 in Ivanova et al.).
> A primary contribution of their works is the design of the different experiment conditions to disambiguate code comprehension from natural language comprehension, and their key result shows that brain regions do respond to code selectively after controlling for various language-like confounds which codes may exhibit.
>
> Further, Ivanova et al., 2020 do not find any group differences in the activations in response to `control flow` or `data type` properties across individuals (Figure 4 in their work).
>
> It is quite possible that a group difference in activations seen in, say, the MD regions is driven by general attentional demands of comprehending code, and not information about the specific computations and constructs required to process code.
> We provide evidence that these regions encode specific program-related properties as well.
> The results from Experiment 1 helps us conclude that the neural activations pertaining to different programming-constructs like loops and datatypes are _different_ enough between programs comprehended by an individual that it is possible to decode this information from neural activations alone.
>
> **An investigation of individual-level code properties encoded in neural activations across different brain regions has not been previously investigated with the detail our experiment design affords, and we provide the first evidence for the properties indeed being encoded.**.
>
> **Proposed draft edit 2 :** We will ensure to highlight this difference in the introduction and discussion sections in our current draft.
>
> ***
>
> ### Significance of reported results
> > Control flow and data type: The authors make claims about accuracy differences between MD & LS. However, these differences are not statistically significant by their own admission
>
> > I found weak evidence for the identification accuracies presented here since several of the tests were not statistically significant and did not beat the random embedding baseline.
>
> We highlight that in both our experiments, we perform two sets of analyses --
> 1. Main analysis - Whether the code properties (Exp 1) or code models (Exp 2) can be decoded from brain activations.
> 2. Additional analysis - Whether any one brain region preferentially encodes any of the code properties or code models.
>
> **The key research question in Exp 1 is whether brain activations even encode code properties, and the key research question in Exp 2 is whether it is even possible to sufficiently decode code model representations.
> We find statistically robust results for both these key questions**.
>
> Only in our analyses of comparing different brain regions, in looking for whether any one brain system encodes code properties preferentially (Exp 1) or whether any one code model architecture aligns best to the information stored in any one brain system (Exp 2), do we find some instances of weak evidence.
>
> That said, we nevertheless gather valuable insights from these comparisons.
> - We observe that MD decodes representations from 3 models significantly better than LS, and 4 models significantly better than Vis.
> These data provide evidence that a collection of code models (albeit not all) map more closely to MD than other regions, which is significant and of relevance.
> - Further, in Experiment 1, MD and LS decode `Control Flow` better than Vis, MD decodes `Dynamic Analysis` better than Vis, and LS decodes `Static Analysis` better than Vis.
> These results suggest that the MD and LS systems definitely encode code-specific properties which go beyond what can be discerned from the visual layout of the code.
> - In Experiment 2, we confirm that these encodings are likely driven by token-level information (details mentioned in the following comment `Motivation for Experiment 2`).
> We provide the results of all statistical comparisons in tables 8-11, Appendix E.
>
> The fact that we do not find subsets of brain regions preferentially encoding specific subsets of properties or models should not invalidate these key findings and insights we learn.

---

> ### Author Response · Authors · 2021-11-17
> **Response to reviewer vpX5 - Part 3**
>
>
>
> Further, the activity in the MD, LS, Vis brain regions corresponding to the remaining four properties that we explore in this work--`code vs sentences`, `variable language`, `data type`, and `control flow` cannot be attributed to program length, since the lengths of the programs were controlled to be the same to a large extent across these categories in the original dataset by Ivanova et al.
>
> To confirm this, we had looked at the correlation of program length (`# of tokens`) to the metrics which were significantly decoded (Figure 3)--`control flow` and the `data type` used in the programs (`code vs. sentences` does not lend itself to a correlation analysis since the comparison there is done between English sentences and programs).
> These are documented in the first three rows of Table 17---`Datatype` refers to math vs string programs (labelled as 1-0 and then correlated with program length); `Conditional` refers to if-programs vs. the rest; `Iteration` refers to loop-programs vs. the rest.
>
> It is clear from Table 17, Appendix G, that these other metrics are not correlated to program length.
> It is hence unlikely that program length as a confound affects the results we report in Experiment 1 of our work.
>
> > Overall, I am not convinced that classification or identification accuracy is useful in gauging what properties of code are encoded by different brain regions. It would be useful to introduce more contrasts or control for ... The authors are making definitive statements about the observed performance patterns based on prior knowledge of the MD system, LS system etc. However, given the possibility of confounds and correlations between the metrics itself, I am unable to view these results as strong evidence for one mechanism over the other.
>
> As mentioned in our point above, we did evaluate the correlation of each of the ordinal-valued metrics we explore in our work.
> We did find most of these metrics (except runtime steps taken to execute the program) to be correlated to program length.
> This however does not affect the other decodability results that we report in our work, e.g. on being able to distinguish a program containing a loop vs. a conditional statement, strings vs. integer-based operations, and programs from sentences describing those programs.
>
> **These are the key results and insights we wish to convey in Experiment 1.
> The fact that we can distinguish different control flow and data types from brain-activity from reading a program alone is a key result. We further confirm that these observations are not correlated to expected confounds like program length, suggesting that we are able to distinguish more than just program lengths from brain activity of comprehended code.**
>
> If the reviewer has suggestions for other concrete program metrics which will help disambiguate representations of finer properties of programs, we would love to consider them.
>
> Addressing a related minor clarification question you had raised -
> > Since the magnitude of the predicted value is important here, I am confused by the use of linear correlation as opposed to RMSE.
>
> We use correlations to compute the decoding results of the static and dynamic analysis metrics (ordinal-valued) in order to ease comprehension. Having correlations as a metric establishes 0 as the random baseline, which makes it easier to comprehend the results in Figure 3.
> We agree that a natural distance metric to use here is the RMSE, and we had looked at RMSE as well when analyzing our results, and had found the trend to be the same as correlations.
>
> A summary of RMSE results --
>
> | Brain Network | Code Property | RMSE | Null RMSE Mean | Null RMSE SD | Significant? (p<0.01) |
> |---------------|---------------|------|----------------|--------------|--------------|
> | MD | Static analysis | 7.67 | 8.27 | 0.15 | Yes |
> | MD | Dynamic analysis | 3.48 | 4.03 | 0.07 | Yes |
> | LS | Static analysis | 7.33 | 8.00 | 0.14 | Yes |
> | LS | Dynamic analysis | 3.67 | 3.95 | 0.07 | Yes |
> | Vis | Static analysis | 7.99 | 8.27 | 0.15 | No |
> | Vis | Dynamic analysis | 3.79 | 4.04 | 0.08 | Yes |
> | Aud | Static analysis | 8.04 | 8.10 | 0.15 | No |
> | Aud | Dynamic analysis | 3.79 | 3.97 | 0.07 | No |
>
> **Proposed draft edit 1 :** We will append these RMSE results to the appendix - we had omitted them in the submitted draft.
> ***

---

> ### Author Response · Authors · 2021-11-17
> **Response to reviewer vpX5 - Part 2**
>
>
>
> ### Program length as a potential confound
> > The authors note that the visual ROI can effectively classify 4 different stimulus properties but it does not significantly predict the token count. This to me suggests that the 4 properties evaluated here are in fact strongly confounded with program length, presence of letters etc. and may not specifically correspond to code comprehension. What do the authors think about such confounds being relevant in other areas as well and its impact on the conclusions reached in the paper?
>
> The six properties we evaluate -- `code vs sentences`, `variable language`, `data type`, `control flow`, `static analysis` properties, `dynamic analysis` properties, are not all correlated to program length.
> We have documented these inter-correlations in Appendix G of the submitted draft.
> We had suspected program length to be a key potential confound and had hence performed the correlation analysis that is documented in Appendix G, Table 17.
> As we mention in our draft, we only found the `static analysis` metrics (`Halstead complexity`, `Cyclomatic complexity`, `# of AST nodes` in a program, `# of bytecode instructions executed`) to be correlated with program length (`# of tokens` in a program).
> We hence evaluate just one of these static analysis metrics (token count) in all our core analyses instead of considering all the metrics separately.

---

> > ### Comment · Reviewer_vpX5 · 2021-11-20
> > **Response Part 1**
> >
> > Thank you for considering my comments and the responses!
> >
> > The 	point I was raising here is that visual cortex itself can effectively classify several of these properties (4/6 are significant in Fig 3). Obviously, visual cortex is not doing high-level processing of code and instead cares about visual presentations features. Taken together, this suggests that the features considered here have sufficient low-level information. How then does the current approach distinguish between ROI1 that does well because of low-levels features vs. ROI2 that also does well but not because of low-level features?
> >
> > Imo it cannot do this well but by variance partitioning you could have identified the unique variance captured by say “control flow” that is not explained by visual features.
> >
> > PS From table 17, it seems to me that everything but "iteration" and "runtime steps" is reasonably correlated with program length ($r>0.4$)- enough that I would want to know the variance explained by these variables uniquely.
> >
> > Thank you for adding the RMSE results, they are indeed identical!

---

> > > ### Author Response · Authors · 2021-11-22
> > > **Measuring the effect of the visual system**
> > >
> > > Thanks for the follow-up.
> > >
> > > > Imo it cannot do this well but by variance partitioning you could have identified the unique variance captured by say “control flow” that is not explained by visual features.
> > >
> > > Thank you for this useful suggestion.
> > >
> > > On using variance partitioning (VP) -- we note that our current study is in the decoding setting, where we're decoding these specific code properties from brain response data. VP on the other hand is applicable in the encoding setting, where the research question addressed will not be the same as our current formulation. In our study, the focus is on investigating what properties a multivariate representation of brain activity encodes.
> > >
> > > That said, to address the valid concern of subtracting the effect of the activity from the visual system, we tried the following experiment -
> > > - we learn two models --
> > >    - $M_1$, a joint-model, which decodes from multiple regions, say $R_1$ and $R_2$
> > >    - $M_2$, which decodes from a single region $R_2$.
> > > - If the difference in predictions between $M_1$ and $M_2$ is positive and significant, we conclude that the region $R_1$ has at least some information orthogonal to $R_2$ which contributes to the increased difference in decoding accuracy.
> > >
> > > The table below shows the key results from this experiment.
> > >
> > >
> > >
> > > |Code Property | Regions in $M_1$ | Regions in $M_2$ | t | Is significant?|
> > > |--------------|----------------|----------------|---|----------------|
> > > |Code vs. Sentence |  LS+Vis |   LS |   3.88 | 1 |
> > > |Code vs. Sentence |  LS+Vis |   Vis |   3.56 | 1 |
> > > |Code vs. Sentence |  MD+LS |    LS |   3.77 | 1 |
> > > |Code vs. Sentence |  MD+LS |    MD |   3.83 | 1 |
> > > |Code vs. Sentence |  MD+Vis |   MD |   3.25 | 1 |
> > > |Code vs. Sentence |  MD+Vis |   Vis |   2.96 | 1 |
> > > |Control Flow      |  LS+Vis |   Vis |   2.91 | 1 |
> > > |Control Flow      |  MD+Vis |   Vis |   4.11 | 1 |
> > > |Data Type         |  MD+Vis |   Vis |   2.86 | 1 |
> > >
> > >
> > > This suggests that the MD system does encode information which is different from the visual system for the three properties -- control flow (presence of loops and if statements), data type (string vs numeric operations), and in processing code vs. sentences (reading a program vs reading text describing it).
> > >
> > > The two other properties - static and dynamic analysis metrics, as suggested by their high inter-correlation and correlation to program length, cannot be distinguished any better from MD or LS response activity than Vis response acitivity.
> > >
> > > We have added a subsection *Analysis 3* in Section 5.1, which reflects this discussion.
> > > Detailed results are available in Table 15, Appendix F.

---

> ### Author Response · Authors · 2021-11-17
> **Response to reviewer vpX5 - Part 1**
>
>
> Thank you for your detailed review of our work!
> We're glad you found the questions we explore in this work interesting.
> Your thoughtful comments have definitely helped us sharpen the arguments we present in this work.
> Many of the comments you provide can in fact be addressed by clarifying our overall framing of the results.
> We address the key points below and will update the manuscript to provide a clearer context for our findings.
>
> ### A summary of key points of feedback and our responses
> The main points of feedback we gather from your comments are -
> 1. The program properties we test may all be inter-correlated, and in turn be correlated to the length of a program.
> By not accounting for this confound, our claims may not be valid.
> 2. In Experiment 1, it is not clear how our results are any different from the key findings of Liu et al., 2020 and Ivanova et al., 2020.
> 3. It is unclear how to interpret some of the results in both Experiments 1 and 2, given how some comparisons reported are not statistically significant.
> 4. The motivation behind setting up Experiment 2 is unclear.
>
>
> We respond to these important questions below in detail in separate sections.
> A summary of our responses are -
> 1. We provide details of an inter-correlation analysis in Appendix G.
> This analysis suggests that program length is not a direct confound affecting all the code properties we test.
> 2. Experiment 1 helps us conclude that the neural activations pertaining to different programming-constructs like loops and datatypes are _different_ enough between programs comprehended by an individual that it is possible to decode this information from neural activations alone.
> Liu et al., 2020 and Ivanova et al., 2020 instead investigate the regions of the brain (MD, LS) where activations to all comprehended code are observed _across individuals_ (Figure 3 in Liu et al. and Figure 2 in Ivanova et al.).
> In fact, Ivanova et al., 2020 do not find any group differences in the activations in response to `control flow` or `data type` properties across individuals (Figure 4 in their work).
> It is possible that a group difference in activations seen in, say, the MD regions is driven by general attentional demands and not information about the exact computations required to process the code.
> We provide evidence that these regions encode specific program-related properties as well.
> 3. In both Experiment 1 and 2, we clarify that some tests of _differences_ in key experiment conditions end up being insignificant.
> The key research questions we wish to answer in this work do not pertain to studying these differences, but rather asserting the presence of specific code-related properties in brain representations of code.
> All the key results in Experiment 1 are statistically robust.
> In Experiment 2, when specifically comparing models of different complexity (one of the key questions we want to answer), we agree that we do not have statistically robust results to back the general trends we see, but do see a strong numerical trend. We will modify our draft to reflect this clearly.
> 4. Through Experiment 2, we want to understand whether information encoded by extant code models reflects the neural activity pertaining to code comprehension.
> From our current results, we conclude that the brain systems at least significantly encode token-level information, and explain the correspondence we see with representations of code models.
>
> ***
>
> Detailed answers to your points of feedback and questions follow --

---

### Decision · Program_Chairs · 2022-01-20

**Decision:**

Reject

**Comment:**

This paper aims to relate brain activity (of people reading computer
code) to properties of the computer code. They relate the found
representations to those obtained from ML computational language
models applied to the same programs.  The paper is clearly written and
an interesting idea.

There was a lot of discussion and the author(s) updated their paper a
lot.  Program length as a potential confound was raised and
successfully rebutted.  The extent of novelty from Ivanova et al 2020
was also discussed and successfully rebutted.  In the end, the main
issues the reviewers had were 1) that the paper had been updated
substantially since submission (and would therefore benefit from a
thorough re-review) and 2) whether the results provide enough new
insights about the brain or about ML language models.

To summarize, the authors spent a lot of time addressing issues in the
rebuttal phase and the paper got a lot better with the reviewers'
suggestions, but reviewers agreed it would benefit from more work and
further review before acceptance.  I agree with this assessment.